# The Bayesian Stability Zoo

**Shay Moran**
Department of Mathematics
& Department of Computer Science
Technion – Israel Institute of Technology;
smoran@technion.ac.il

**Hilla Schefler**
Department of Mathematics
Technion – Israel Institute of Technology
hillas@campus.technion.ac.il

**Jonathan Shafer**
Computer Science Division
UC Berkeley
shaferjo@berkeley.edu

## Abstract

We show that many definitions of stability found in the learning theory literature are equivalent to one another. We distinguish between two families of definitions of stability: *distribution-dependent* and *distribution-independent Bayesian stability*. Within each family, we establish equivalences between various definitions, encompassing approximate differential privacy, pure differential privacy, replicability, global stability, perfect generalization, TV stability, mutual information stability, KL-divergence stability, and Rényi-divergence stability. Along the way, we prove boosting results that enable the amplification of the stability of a learning rule. This work is a step towards a more systematic taxonomy of stability notions in learning theory, which can promote clarity and an improved understanding of an array of stability concepts that have emerged in recent years.

## 1   Introduction

Algorithmic stability is a major theme in learning theory, where seminal results have firmly established its close relationship with generalization. Recent research has further highlighted the intricate interplay between stability and additional properties of interest beyond statistical generalization. These properties encompass privacy [DMNS06], fairness [HKRR18], replicability [BGH+23, ILPS22], adaptive data analysis [DFH+15b, DFH+15a], and mistake bounds in online learning [ALMM19, BLM20].

This progress has come with a proliferation of formal definitions of stability, including pure and approximate Differential Privacy [DMNS06, DKM+06], Perfect Generalization [CLN+16], Global Stability [BLM20], KL-Stability [McA99], TV-Stability [KKMV23], $f$-Divergence Stability [EGI20], Rényi Divergence Stability [EGI20], and Mutual Information Stability [XR17, BMN+18], as well as related combinatorial quantities such as the Littlestone dimension [Lit87] and the clique dimension [AMSY23].

It is natural to wonder to what extent these various and sundry notions of stability actually differ from one another. The type of equivalence we consider between definitions of stability is as follows.

37th Conference on Neural Information Processing Systems (NeurIPS 2023).

> *Definition A and Definition B are **weakly equivalent** if for every hypothesis class $\mathcal{H}$ the following holds:*
>
> | $\mathcal{H}$ has a PAC learning rule that is stable according to Definition A | $\iff$ | $\mathcal{H}$ has a PAC learning rule that is stable according to Definition B |

This type of equivalence is weak because it does *not* imply that a learning rule satisfying one definition also satisfies the other.

Recent results show that many stability notions appearing in the literature are in fact weakly equivalent. The work of [BGH$^+$23] has shown sample efficient reductions between approximate differential privacy, replicability, and perfect generalization. Combined with the work of [ABL$^+$22, ILPS22, KKMV23, MM22], a rich web of equivalences is being uncovered between approximate differential privacy and other definitions of algorithmic stability (see Fig. 1).

In this paper we extend the study of equivalences between notions of stability, and make it more systematic. Our starting point is the following observation: many of the definitions mentioned above belong to a broad family of definitions of stability, which we informally call *Bayesian definitions of stability*. Definitions in this family roughly take the following form: a learning rule $A$ is considered stable if the quantity

$$d\Big(A(S), \mathcal{P}\Big)$$

is small enough, where:

- $d$ is a measure of dissimilarity between distributions.

- $\mathcal{P}$ is a specific *prior distribution* over hypotheses;

- $A(S)$ is the *posterior distribution*, i.e., the distribution of hypotheses generated by the learning rule $A$ when applied to the input sample $S$.

Namely, a Bayesian definition of stability is parameterized by a choice of $d$, a choice of $\mathcal{P}$, and a specification of how small the dissimilarity is required to be.[1]

**Remark 1.1.** *To understand our choice of the name* Bayesian *stability, recall that the terms* prior *and* posterior *come from Bayesian statistics. In Bayesian statistics the analyst has some prior distribution over possible hypothesis before conducting the analysis, and chooses a posterior distribution over hypotheses when the analysis is complete. Bayesian stability is defined in terms of the dissimilarity between these two distributions.*

A central insight of this paper is that there exists a meaningful distinction between two types of Bayesian definitions, based on whether the choice of the prior $\mathcal{P}$ depends on the population distribution $\mathcal{D}$:

- Distribution-*independent* (DI) stability. These are Bayesian definitions of stability in which $\mathcal{P}$ is some fixed prior that depends only on the class $\mathcal{H}$ and the learning rule $A$, and does not depend on the population distribution $\mathcal{D}$. Namely, they take the form:

$$\exists \text{ prior } \mathcal{P} \; \forall \text{ population } \mathcal{D} \; \forall m \in \mathbb{N} : \; d(A(S), \mathcal{P}) \text{ is small},$$

  where $S \sim \mathcal{D}^m$.

- Distribution-*dependent* (DD) stability. Here, the prior may depend also on $\mathcal{D}$, so each population distribution $\mathcal{D}$ might have a different prior. Namely:

$$\forall \text{ population } \mathcal{D} \; \exists \text{ prior } \mathcal{P}_\mathcal{D} \; \forall m \in \mathbb{N} : \; d(A(S), \mathcal{P}_\mathcal{D}) \text{ is small}.$$

A substantial body of literature has investigated the interconnections among distribution-dependent definitions. In Theorem 1.4, we provide a comprehensive summary of the established equivalences. A

---

[1] An example for an application in the context of generalization is the classic PAC Bayes Theorem. The theorem assures that for every population distribution and any given prior $\mathcal{P}$, the difference between the population error of an algorithm $A$ and the empirical error of $A$ is bounded by $\tilde{O}\left(\frac{\sqrt{\text{KL}(A(S), \mathcal{P})}}{m}\right)$, where $m$ is the size of the input sample $S$, and the KL divergence is the "measure of dissimilarity" between the prior and the posterior . See e.g. Theorem 3.2.

natural question arises as to whether a similar web of equivalences exists for distribution-independent definitions. Our principal contribution is to affirm that, indeed, such a network exists. Identifying such equivalences is a step towards creating a comprehensive taxonomy of stability definitions.

## 1.1 Our Contribution

Our first main contribution is an equivalence between distribution-independent definitions of stability.

**Theorem** (**Informal Version of Theorem 2.1**). *The following definitions of stability are weakly equivalent:*

1. *Pure Differential Privacy;*                                              *(Definition 3.5)*

2. *Distribution-Independent* KL-*Stability;*                             *(Definition 3.6)*

3. *Distribution-Independent One-Way Pure Perfect Generalization;*      *(Definition 3.7)*

4. *Distribution-Independent* $D_\alpha$-*Stability for* $\alpha \in (1, \infty)$.            *(Definition 3.6)*

*Where* $D_\alpha$ *is the Rényi divergence of order* $\alpha$. *Furthermore, a hypothesis class* $\mathcal{H}$ *has a PAC learning rule that is stable according to one of these definitions if and only if* $\mathcal{H}$ *has finite fractional clique dimension (See Appendix B.1).*

**Remark 1.2.** *Observe that DI* KL-*stability is equivalent to DI* $D_1$-*stability, and DI one-way pure perfect generalization is equivalent to DI* $D_\infty$-*stability. Therefore, The above theorem can be viewed as stating a weak equivalence between pure differential privacy and* $D_\alpha$-*stability for* $\alpha \in [1, \infty]$.

**Remark 1.3.** *In this paper we focus purely on the information-theoretic aspects of learning under stability constraints, and therefore we consider learning rules that are mathematical functions, and disregard considerations of computability and computational complexity.*

Table 1 summarizes the distribution-independent definitions discussed in Theorem 2.1. All the definitions in each row are weakly equivalent.

Table 1: Distribution-independent Bayesian definitions of stability.

| Name | Dissimilarity | Definition |
|---|---|---|
| KL-Stability | $\mathbb{P}_S[\mathsf{KL}(A(S) \parallel \mathcal{P}) \leq o(m)] \geq 1 - o(1)$ | 3.6 |
| $D_\alpha$-Stability | $\mathbb{P}_S[\mathsf{D}_\alpha(A(S) \parallel \mathcal{P}) \leq o(m)] \geq 1 - o(1)$ | 3.6 |
| Pure Perfect Generalization | $\mathbb{P}_S[\forall \mathcal{O} : A(S)(\mathcal{O}) \leq e^{o(m)} \mathcal{P}(\mathcal{O})] \geq 1 - o(1)$ | 3.7 |

One example for how the equivalence results can help build bridges between different stability notions in the literature is the connection between pure differential privacy and the PAC-Bayes theorem. Both of these are fundamental ideas that have been extensively studied. Theorem 2.1 states that a hypothesis class admits a pure differentially private PAC learner if and only if it admits a distribution independent KL-stable PAC learner. This is an interesting and non-trivial connection between two well studied notions. As a concrete example of this connection, recall that thresholds over the real line cannot be learned by a differentially private learner [ALMM19]. Hence, by Theorem 2.1, there does not exist a PAC learner for thresholds that is KL-stable. Another example is half-spaces with margins in $\mathbb{R}^d$. Half-spaces with margins are differentially private learnable [?], therefore there exists a PAC learner for half-spaces with margins that is KL-stable.

Our second main contribution is a boosting result for weak learners that have bounded KL-divergence with respect to a distribution-independent prior. Our result demonstrates that distribution-independent KL-stability is boostable. It is interesting to see that one can simultaneously boost both the stability and the learning parameters of an algorithm.

**Theorem** (**Informal Version of Theorem 2.2**). *Let* $\mathcal{H}$ *be a hypothesis class. If there exists a weak learner A for* $\mathcal{H}$, *and there exists a prior distribution* $\mathcal{P}$ *such that the expectation of* $\mathsf{KL}(A(S) \parallel \mathcal{P})$ *is bounded, then there exists a* KL-*stable PAC learner that admits a logarithmic divergence bound.*

The proof of Theorem 2.2 relies on connections between boosting of PAC learners and online learning with expert advice.

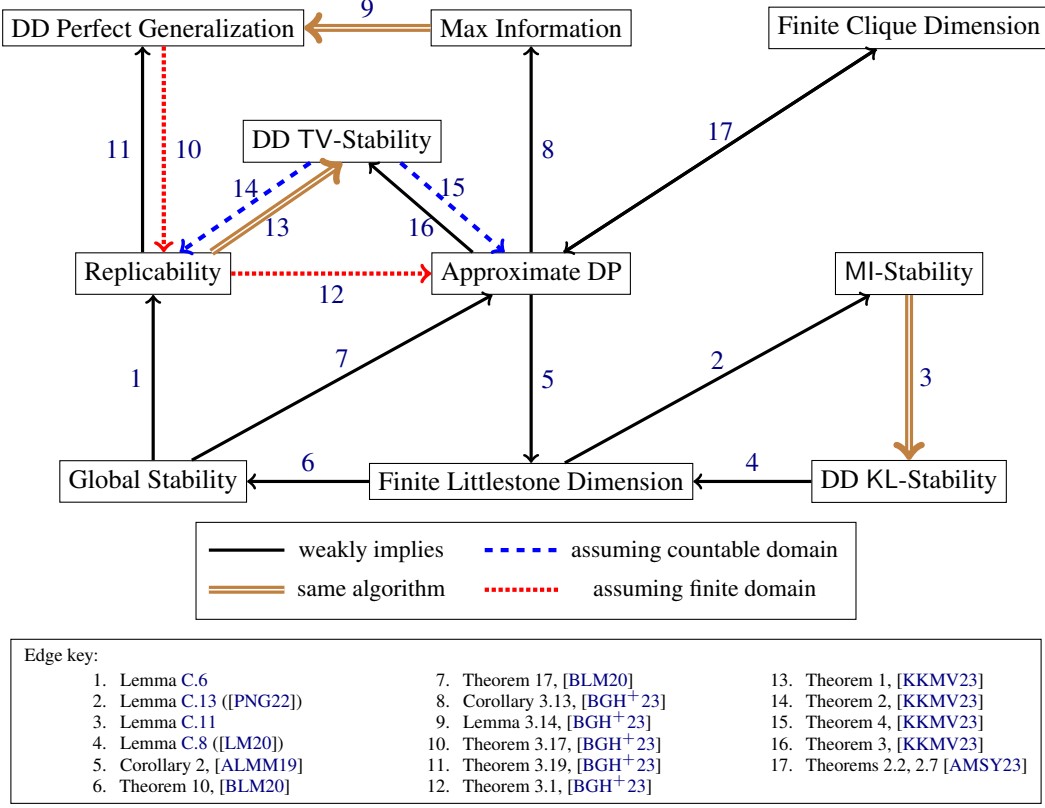

Figure 1: A summary of equivalences between distribution-dependent definitions of stability (Theorem 1.4). A solid black arrow from $A$ to $B$ means that definition $A$ weakly implies definition $B$. A dashed blue arrow from $A$ to $B$ means that $A$ weakly implies $B$ only if the domain $\mathcal{X}$ is countable. A dotted red arrow from $A$ to $B$ means that $A$ weakly implies $B$ only if the domain $\mathcal{X}$ is finite. A double brown arrow from $A$ to $B$ means that every learning rule that satisfies definition $A$ also satisfies definition $B$.

Lastly, after conducting an extensive review of the literature, we have compiled a comprehensive network of equivalence results for distribution-dependent definitions of stability. This network is presented in Theorem 1.4, Figure 1, and Table 2.

**Theorem 1.4** (Distribution-Dependent Equivalences; [ABL+22, ILPS22, MM22, PNG22, BGH+23, KKMV23]). *The following definitions of stability are weakly equivalent with respect to an arbitrary hypothesis class $\mathcal{H}$:*

1. *Approximate Differential Privacy;*          (*Definition 3.5*)

2. *Distribution-Dependent* KL-*Stability;*          (*Definition 3.6*)

3. *Mutual-Information Stability;*          (*Definition 3.12*)

4. *Global Stability.*          (*Definition 3.11*)

*If the domain is countable then the following are also weakly equivalent to the above:*

5. *Distribution-Dependent* TV-*Stability;*          (*Definition 3.13*)

6. *Replicability.*          (*Definition 3.8*)

*If the domain is finite then the following are also weakly equivalent to the above:*

7. *One-Way Perfect Generalization;*          (*Definition 3.7*)

8. *Max Information.*          (*Definition 3.14*)

*Furthermore, for any hypothesis class $\mathcal{H}$, the following conditions are equivalent:*

- $\mathcal{H}$ has a PAC learning rule that is stable according to one of the definitions *1* to *6* (and the cardinality of the domain is as described above);

- $\mathcal{H}$ has finite Littlestone dimension;                                    *(Definition C.3)*

- $\mathcal{H}$ has finite clique dimension.                                         *(Definition C.5)*

We emphasize that Theorem 1.4 is a summary of existing results, and is not a new result. We believe that our compilation serves as a valuable resource, and that stating these results here in a unified framework helps to convey the conceptual message of this paper. Namely, the fact that a large number of disparate results can neatly be organized based on our notions of distribution-dependent and distribution-independent definitions of stability is a valuable observation that can help researchers make sense of the stability landscape.

Table 2: Distribution-dependent Bayesian definitions of stability.

| Name | Dissimilarity | Definition | References |
|---|---|---|---|
| KL-Stability | $\mathbb{P}_S[\mathsf{KL}(A(S) \parallel \mathcal{P}_{\mathcal{D}}) \leq o(m)] \geq 1 - o(1)$ | 3.6 | [McA99] |
| TV-Stability | $\mathbb{E}_S[\mathsf{TV}(A(S), \mathcal{P}_{\mathcal{D}})] \leq o(1)$ | 3.13 | [KKMV23] |
| MI-Stability | $\mathbb{E}_S[\mathsf{KL}(A(S) \parallel \mathcal{P}_{\mathcal{D}})] \leq o(m)$ | 3.12 | [XR17, BMN+18] |
| Perfect Generalization | $\mathbb{P}_S[\forall \mathcal{O} : A(S)(\mathcal{O}) \leq e^{\varepsilon}\mathcal{P}_{\mathcal{D}}(\mathcal{O}) + \delta] \geq 1 - o(1)$ | 3.7 | [CLN+16] |
| Global Stability | $\mathbb{P}_{S,h\sim\mathcal{P}_{\mathcal{D}}}[A(S) = h] \geq \eta$ | 3.11 | [BLM20] |
| Replicability | $\mathbb{P}_{r\sim\mathcal{R}}\left[\mathbb{P}_{S,h_r\sim\mathcal{P}_{\mathcal{D},r}}[A(S; r) = h_r] \geq \eta\right] \geq \nu$ | 3.10 | [BGH+23, ILPS22] |

## 1.2 Related Works

The literature on stability is vast. Stability has been studied in the context of optimization, statistical estimation, regularization (e.g., [Tik43] and [Phi62]), the bias-variance tradeoff, algorithmic stability (e.g., [BE02]; see bibliography in Section 13.6 of [SB14]), bagging [Bre96], online learning and optimization and bandit algorithms (e.g., [Han58]; see bibliography in Section 28.6 of [LS20]), and other topics.

There are numerous definitions of stability, including pure and approximate Differential Privacy [DMNS06, DKM+06], Perfect Generalization [CLN+16], Global Stability [BLM20], KL-Stability [McA99], TV-Stability [KKMV23], $f$-Divergence Stability [EGI20], Rényi Divergence Stability [EGI20], and Mutual Information Stability [XR17, BMN+18].

Our work is most directly related to the recent publication by Bun et al. [BGH+23]. They established connections and separations between replicability, approximate differential privacy, max-information and perfect generalization for a broad class of statistical tasks. The reductions they present are sample-efficient, and nearly all are computationally efficient and apply to a general outcome space. Their results are central to the understanding of equivalences between notions of stability as laid out in the current paper.

A concurrent work by Kalavasis et al. [KKMV23] showed that TV-stability, replicability and approximate differential privacy are equivalent; this holds for general statistical tasks on countable domains, and for PAC learning on any domain. They also provide a statistical amplification and TV-stability boosting algorithm for PAC learning on countable domains.

Additionally, recent works [AUZ23, HKMN23] have shown an equivalence between differential privacy and robustness for estimation tasks.

Theorem 2.2 is a boosting result. Boosting has been a central topic of study in computational learning theory since its inception in the 1990s by Schapire [**?**] and Freund [Fre95]. The best-known boosting algorithm is AdaBoost [FS97], which has been extensively studied. Boosting also has rich connections with other topics such as game theory, online learning, and convex optimization (see [SF12], Chapter 10 in [SB14], and Chapter 7 in [MRT18]).

## 2 Technical Overview

This section presents the complete versions of Theorems 1.4 and 2.2. We provide a concise overview of the key ideas and techniques employed in the proofs. All proofs appear in the appendices.

Please refer to Section 3 for a complete overview of preliminaries, including all technical terms and definitions.

## 2.1 Equivalences between DI Bayesian Notions of Stability

The following theorem, which is one of the main results of this paper, shows the equivalence between different distribution-independent definitions. The content of Theorem 2.1 is summarized in Table 1.

**Theorem 2.1** (Distribution-Independent Equivalences). *Let $\mathcal{H}$ be a hypothesis class. The following is equivalent.*

1. *There exists a learning rule that PAC learns $\mathcal{H}$ and satisfied pure differential privacy (Definition 3.5).*

2. *$\mathcal{H}$ has finite fractional clique dimension.*

3. *For every $\alpha \in [1, \infty]$, there exists a learning rule that PAC learns $\mathcal{H}$ and satisfied distribution-independent $\mathsf{D}_\alpha$-stability (Definition 3.6).*

4. *For every $\alpha \in [1, \infty]$, there exists a distribution-independent $\mathsf{D}_\alpha$-stable PAC learner $A$ for $\mathcal{H}$, that satisfies the following:*

   (i) *$A$ is interpolating almost surely. Namely, for every $\mathcal{H}$-realizable distribution $\mathcal{D}$, $\mathbb{P}_{S \sim \mathcal{D}^m}[\mathrm{L}_S(A(S)) = 0] = 1$.*

   (ii) *$A$ admits a divergence bound of $f(m) = O(\log m)$, with confidence $\beta(m) \equiv 0$. I.e., for every $\mathcal{H}$-realizable distribution $\mathcal{D}$, $\mathsf{D}_\alpha(A(S) \| \mathcal{P}) \leq O(\log m)$ with probability 1, where $S \sim \mathcal{D}^m$ and $\mathcal{P}$ is a prior distribution independent of $\mathcal{D}$.*

   (iii) *For every $\mathcal{H}$-realizable distribution $\mathcal{D}$, the expected population loss of $A$ with respect to $\mathcal{D}$ satisfies $\mathbb{E}_{S \sim \mathcal{D}^m}[\mathrm{L}_\mathcal{D}(A(S))] \leq O\left(\sqrt{m^{-1} \log m}\right)$.*

*In particular, plugging $\alpha = 1$ in Item (ii) implies $\mathsf{KL}$-stability with divergence bound of $f(m) = O(\log m)$ and confidence $\beta(m) \equiv 0$. Plugging $\alpha = \infty$ implies distribution-independent one-way $\varepsilon$-pure perfect generalization, with $\varepsilon(m) \leq O(\log m)$ and confidence $\beta(m) \equiv 0$.*

### 2.1.1 Proof Idea for Theorem 2.1

We prove the following chain of implications:

$$\text{Pure DP} \xoverset{(1)}{\Longrightarrow} \mathsf{D}_\infty\text{-Stability} \xoverset{(2)}{\Longrightarrow} \mathsf{D}_\alpha\text{-Stability } \forall \alpha \in [1, \infty] \xoverset{(3)}{\Longrightarrow} \text{Pure DP.}$$

**Pure DP $\implies \mathsf{D}_\infty$-Stability.** The first step towards proving implication (1) is to define a suitable prior distribution $\mathcal{P}$ over hypotheses. The key tool we used in order to define $\mathcal{P}$ is the characterization of pure DP via the fractional clique dimension [AMSY23]. In a nutshell, [AMSY23] proved that (i) a class $\mathcal{H}$ is pure DP learnable if and only if the fractional clique dimension of $\mathcal{H}$ is finite; (ii) the fractional clique dimension is finite if and only if there exists a polynomial $q(m)$ and a distribution over hypothesis $\mathcal{P}_m$, such that for every realizable sample $S$ of size $m$, we have

$$\mathbb{P}_{h \sim \mathcal{P}_m}[\mathrm{L}_S(h) = 0] \geq \frac{1}{q(m)}. \tag{1}$$

(For more details please refer to Appendix B.1.) Now, the desired prior distribution $\mathcal{P}$ is defined to be a mixture of all the $\mathcal{P}_m$'s.

The next step in the proof is to define a learning rule $A$: (i) sample hypotheses from the prior $\mathcal{P}$; (ii) return the first hypothesis $h$ that is consistent with the input sample $S$ (i.e. $\mathrm{L}_S(h) = 0$). $A$ is well-defined since with high probability it will stop and return a hypothesis after $\approx q(m)$ re-samples from $\mathcal{P}$. Since the posterior $A(S)$ is supported on $\{h : \mathrm{L}_S(h) = 0\}$, a simple calculation which follows from Equation (1) shows that for every realizable distribution $\mathcal{D}$, $\mathsf{D}_\infty(A(S) \| \mathcal{P}) \leq \log(q(m))$ almost surly where $S \sim \mathcal{D}^m$.

Finally, since for $\alpha \in [1, \infty]$ the Rényi divergence $\mathsf{D}_\alpha(\mathcal{Q}_1 \| \mathcal{Q}_2)$ is non-decreasing in $\alpha$ (see Lemma A.1), we conclude that $\mathsf{KL}(A(S) \| \mathcal{P}) \leq O(\log m)$, hence by PAC-Bayes theorem $A$ generalizes.

**$\mathsf{D}_\infty$-Stability $\implies \mathsf{D}_\alpha$-Stability $\forall \alpha \in [1, \infty]$.** This implication is immediate since the Rényi divergence $\mathsf{D}_\alpha(\mathcal{Q}_1 \| \mathcal{Q}_2)$ is non-decreasing in $\alpha$.

$D_\alpha$-**Stability** $\forall \alpha \in [1, \infty] \implies$ **Pure DP.** In fact, it suffices to assume KL-stability. We prove that the promised prior $\mathcal{P}$ satisfies that for every realizable sample $S$ of size $m$, we have $\mathbb{P}_{h \sim \mathcal{P}}[\mathrm{L}_S(h) = 0] \geq \frac{1}{\mathsf{poly}(m)}$, and conclude that $\mathcal{H}$ is pure DP learnable. Given a realizable sample $S$ of size $m$, we uniformly sample $\approx m \log m$ examples from $S$ and feed the new sample $S'$ to the promised KL-stable learner $A$. By noting that if $\mathsf{KL}(A(S') \parallel \mathcal{P})$ is small, one can lower bound the probability of an event according to $\mathcal{P}$ by its probability according to $A(S')$. The proof then follows by applying a standard concentration argument.

## 2.2 Stability Boosting

We prove a boosting result for weak learners with bounded KL with respect to a distribution-independent prior. We show that every learner with bounded KL that slightly beats random guessing can be amplified to a learner with logarithmic KL and expected loss of $O(\sqrt{m^{-1} \log m})$.

**Theorem 2.2** (Boosting Weak Learners with Bounded KL). *Let $\mathcal{X}$ be a set, let $\mathcal{H} \subseteq \{0,1\}^{\mathcal{X}}$ be a hypothesis class, and let $A$ be a learning rule. Assume there exists $k \in \mathbb{N}$ and $\gamma > 0$ such that*

$$\forall \mathcal{D} \in \mathsf{Realizable}(\mathcal{H}) : \ \mathbb{E}_{S \sim \mathcal{D}^k}[\mathrm{L}_{\mathcal{D}}(A(S))] \leq \frac{1}{2} - \gamma, \tag{2}$$

*and there exists $\mathcal{P} \in \Delta(\{0,1\}^{\mathcal{X}})$ and $b \geq 0$ such that*

$$\forall \mathcal{D} \in \mathsf{Realizable}(\mathcal{H}) : \ \mathbb{E}_{S \sim \mathcal{D}^k}[\mathsf{KL}(A(S) \parallel \mathcal{P})] \leq b. \tag{3}$$

*Then, there exists an interpolating learning rule $A^\star$ that PAC learns $\mathcal{H}$ with logarithmic KL-stability. More explicitly, there exists a prior distribution $\mathcal{P}^\star \in \Delta(\{0,1\}^{\mathcal{X}})$ and function $b^\star$ and $\varepsilon^\star$ that depend on $\gamma$ and $b$ such that*

$$\forall \mathcal{D} \in \mathsf{Realizable}(\mathcal{H}) \ \forall m \in \mathbb{N} :$$
$$\mathbb{P}_{S \sim \mathcal{D}^m}[\mathsf{KL}(A^\star(S) \parallel \mathcal{P}^\star) \leq b^\star(m) = O(\log(m))] = 1, \tag{4}$$

$$and$$

$$\mathbb{E}_{S \sim \mathcal{D}^m}[\mathrm{L}_{\mathcal{D}}(A^\star(S))] \leq \varepsilon^\star(m) = O\left(\sqrt{\frac{\log(m)}{m}}\right). \tag{5}$$

### 2.2.1 Proof Idea for Theorem 2.2

The strong learning rule $A^\star$ is obtained by simulating the weak learner $A$ on $O(\log m / \gamma^2)$ samples of constant size $k$ (which are carefully sampled from the original input sample $S$). Then, $A^\star$ returns an aggregated hypothesis – the majority vote of the outputs of $A$. As it turns out, $A^\star$ satisfies logarithmic KL-stability with respect to the prior $\mathcal{P}^\star$ that is a mixture of majority votes of the original prior $\mathcal{P}$. The analysis involves a reduction to regret analysis of online learning using expert advice, and also uses properties of the KL-divergence.

## 3 Preliminaries

### 3.1 Divergences

The Rényi $\alpha$-divergence is a measure of dissimilarity between distributions that generalizes many common dissimilarity measures, including the Bhattacharyya coefficient ($\alpha = 1/2$), the Kullback–Leibler divergence ($\alpha = 1$), the log of the expected ratio ($\alpha = 2$), and the log of the maximum ratio ($\alpha = \infty$).

**Definition 3.1** (Rényi divergence; [Rén61, vEH14]). *Let $\alpha \in (1, \infty)$. The Rényi divergence of order $\alpha$ of the distribution $\mathcal{P}$ from the distribution $\mathcal{Q}$ is*

$$\mathsf{D}_\alpha(\mathcal{P} \parallel \mathcal{Q}) = \frac{1}{\alpha - 1} \log\left(\mathbb{E}_{x \sim \mathcal{P}}\left[\left(\frac{\mathcal{P}(x)}{\mathcal{Q}(x)}\right)^{\alpha - 1}\right]\right).$$

*For $\alpha = 1$ and $\alpha = \infty$ the Rényi divergence is extended by taking a limit. In particular, the limit $\alpha \to 1$ gives the Kullback–Leibler divergence,*

$$\mathsf{D}_1(\mathcal{P} \parallel \mathcal{Q}) = \mathbb{E}_{x \sim \mathcal{P}}\left[\log \frac{\mathcal{P}(x)}{\mathcal{Q}(x)}\right] = \mathsf{KL}(\mathcal{P} \parallel \mathcal{Q}),$$

*and*

$$\mathsf{D}_\infty(\mathcal{P} \parallel \mathcal{Q}) = \log\left(\operatorname*{ess\,sup}_{\mathcal{P}} \frac{\mathcal{P}(x)}{\mathcal{Q}(x)}\right),$$

*with the conventions that $0/0 = 0$ and $x/0 = \infty$ for $x > 0$.*

## 3.2 Learning Theory

We use standard notation from statistical learning (e.g., [SB14]). Given a hypothesis $h : \mathcal{X} \to \{0, 1\}$, the *empirical loss* of $h$ with respect to a sample $S = \{(x_1, y_1), \ldots, (x_m, y_m)\}$ is defined as $\mathrm{L}_S(h) = \frac{1}{m}\sum_{i=1}^m \mathbb{1}[h(x_i) \neq y_i]$. A learning rule $A$ is *interpolating* if for every input sample $S$, $\mathbb{P}_{h \sim A(S)}[\mathrm{L}_S(h) = 0] = 1$. The *population loss* of $h$ with respect to a population distribution $\mathcal{D}$ over $\mathcal{X} \times \{0, 1\}$ is defined as $\mathrm{L}_\mathcal{D}(h) = \mathbb{P}_{(x,y) \sim \mathcal{D}}[h(x) \neq y]$. A population $\mathcal{D}$ over labeled examples is *realizable* with respect to a class $\mathcal{H}$ if $\inf_{h \in \mathcal{H}} \mathrm{L}_\mathcal{D}(h) = 0$. We denote the set of all realizable population distributions of a class $\mathcal{H}$ by $\mathsf{Realizable}(\mathcal{H})$. Given a learning rule $A$ and an input sample $S$ of size $m$, the *population loss* of $A(S)$ with respect to a population $\mathcal{D}$ is defined as $\mathbb{E}_{h \sim A(S)}[\mathrm{L}_\mathcal{D}(h)]$.

A hypothesis class $\mathcal{H}$ is *Probably Approximately Correct (PAC) learnable* if there exists a learning rule $A$ such that for all $\mathcal{D} \in \mathsf{Realizable}(\mathcal{H})$ and for all $m \in \mathbb{N}$, we have $\mathbb{E}_{S \sim \mathcal{D}^m}[\mathrm{L}_\mathcal{D}(A(S))] \leq \varepsilon(m)$, where $\lim_{m \to \infty} \varepsilon(m) = 0$.

**Theorem 3.2** (PAC-Bayes Bound; [McA99, LSM01, McA03]; Theorem 31.1 in [SB14]). *Let $\mathcal{X}$ be a set, let $\mathcal{H} \subseteq \{0, 1\}^\mathcal{X}$, and let $\mathcal{D} \in \Delta(\mathcal{X} \times \{0, 1\})$. For any $\beta \in (0, 1)$ and for any $\mathcal{P} \in \Delta(\mathcal{H})$,*

$$\mathbb{P}_{S \sim \mathcal{D}^m}\left[\forall \mathcal{Q} \in \Delta(\mathcal{H}) : \ \mathrm{L}_\mathcal{D}(\mathcal{Q}) \leq \mathrm{L}_S(\mathcal{Q}) + \sqrt{\frac{\mathsf{KL}(\mathcal{Q} \parallel \mathcal{P}) + \ln(m/\beta)}{2(m-1)}}\right] \geq 1 - \beta.$$

## 3.3 Definitions of Stability

Throughout the following section, let $\mathcal{X}$ be a set called the *domain*, let $\mathcal{H} \subseteq \{0, 1\}^\mathcal{X}$ be a hypothesis class, and let $m \in \mathbb{N}$ be a sample size. A *randomized learning rule*, or a *learning rule* for short, is a function $A : (\mathcal{X} \times \{0, 1\})^* \to \Delta(\{0, 1\}^\mathcal{X})$ that takes a training sample and outputs a distribution over hypotheses. A *population distribution* is a distribution $\mathcal{D} \in \Delta(\mathcal{X} \times \{0, 1\})$ over labeled domain elements, and a *prior distribution* is a distribution $\mathcal{P} \in \Delta(\{0, 1\}^\mathcal{X})$ over hypotheses.

### 3.3.1 Differential Privacy

Differential privacy is a property of an algorithm that guarantees that the output will not reveal any meaningful amount of information about individual people that contributed data to the input (training data) used by the algorithm. See [DR14] for an introduction.

**Definition 3.3.** *Let $\varepsilon, \delta \in \mathbb{R}_{\geq 0}$, and let $\mathcal{P}$ and $\mathcal{Q}$ be two probability measures over a measurable space $(\Omega, \mathcal{F})$. We say that $\mathcal{P}$ and $\mathcal{Q}$ are $(\varepsilon, \delta)$-indistinguishable and write $\mathcal{P} \approx_{\varepsilon, \delta} \mathcal{Q}$, if for every event $\mathcal{O} \in \mathcal{F}$, $\mathcal{P}(\mathcal{O}) \leq e^\varepsilon \cdot \mathcal{Q}(\mathcal{O}) + \delta$ and $\mathcal{Q}(\mathcal{O}) \leq e^\varepsilon \cdot \mathcal{P}(\mathcal{O}) + \delta$.*

**Definition 3.4** (Differential Privacy; [DR14]). *Let $\varepsilon, \delta \in \mathbb{R}_{\geq 0}$. A learning rule $A$ is $(\varepsilon, \delta)$-differentially private if for every pair of training samples $S, S' \in (\mathcal{X} \times \{0, 1\})^m$ that differ on a single example, $A(S)$ and $A(S')$ are $(\varepsilon, \delta)$-indistinguishable.*

Typically, $\varepsilon$ is chosen to be a small constant (e.g., $\varepsilon \leq 0.1$) and $\delta$ is negligible (i.e., $\delta(m) \leq m^{-\omega(1)}$). When $\delta = 0$ we say that $A$ satisfies *pure* differentially privacy.

**Definition 3.5** (Private PAC Learning). *$\mathcal{H}$ is privately learnable or DP learnable if it is PAC learnable by a learning rule $A$ which is $(\varepsilon(m), \delta(m))$-differentially-private, where $\varepsilon(m) \leq 1$ and $\delta(m) = m^{-\omega(1)}$. $A$ is pure DP learnable if the same holds with $\delta(m) = 0$.*

### 3.3.2 $\mathsf{D}_\alpha$-Stability and $\mathsf{KL}$-Stability

**Definition 3.6** ($\mathsf{D}_\alpha$-Stability). *Let $\alpha \in [1, \infty]$. Let $A$ be a learning rule, and let $f : \mathbb{N} \to \mathbb{R}$ and $\beta : \mathbb{N} \to [0, 1]$ satisfy $f(m) = o(m)$ and $\beta(m) = o(1)$.*

1. *A is* distribution-independent $\mathsf{D}_\alpha$-stable *if*

$$\exists \text{ prior } \mathcal{P} \; \forall \text{ population } \mathcal{D} \; \forall m \in \mathbb{N}: \; \mathbb{P}_{S \sim \mathcal{D}^m}[\mathsf{D}_\alpha(A(S) \| \mathcal{P}) \leq f(m)] \geq 1 - \beta(m).$$

2. *A is* distribution-dependent $\mathsf{D}_\alpha$-stable *if*

$$\forall \text{ population } \mathcal{D} \; \exists \text{ prior } \mathcal{P}_\mathcal{D} \; \forall m \in \mathbb{N}: \; \mathbb{P}_{S \sim \mathcal{D}^m}[\mathsf{D}_\alpha(A(S) \| \mathcal{P}_\mathcal{D}) \leq f(m)] \geq 1 - \beta(m).$$

*The function $f$ is called the divergence bound and $\beta$ is called the confidence. The special case of $\alpha = 1$ is referred to as* KL-stability *[McA99].*

### 3.3.3 Perfect Generalization

**Definition 3.7** (One-Way Perfect Generalization). *Let $A$ be a learning rule, and let $\beta : \mathbb{N} \to [0, 1]$ satisfy $\beta(m) = o(1)$.*

1. *Let $\varepsilon : \mathbb{N} \to \mathbb{R}$ satisfy $\varepsilon(m) = o(m)$. A is* $\varepsilon$-pure perfectly generalizing *with confidence $\beta$ if*

$$\exists \text{ prior } \mathcal{P} \; \forall \text{ population } \mathcal{D} \; \forall m \in \mathbb{N}: \; \mathbb{P}_{S \sim \mathcal{D}^m}\Big[\forall \mathcal{O}: \; A(S)(\mathcal{O}) \leq e^{\varepsilon(m)}\mathcal{P}(\mathcal{O})\Big] \geq 1 - \beta(m).$$

2. *([CLN$^+$16]:) Let $\varepsilon, \delta \in \mathbb{R}_{\geq 0}$. A is* $(\varepsilon, \delta)$-approximately perfectly generalizing *with confidence $\beta$ if*

$$\forall \text{ population } \mathcal{D} \; \exists \text{ prior } \mathcal{P}_\mathcal{D} \; \forall m \in \mathbb{N}: \; \mathbb{P}_{S \sim \mathcal{D}^m}[\forall \mathcal{O}: \; A(S)(\mathcal{O}) \leq e^\varepsilon \mathcal{P}_\mathcal{D}(\mathcal{O}) + \delta] \geq 1 - \beta(m).$$

### 3.3.4 Replicability

**Definition 3.8** (Replicability; [BGH$^+$23, ILPS22]). *Let $\rho \in \mathbb{R}_{>0}$ and let $\mathcal{R}$ be a distribution over random strings. A learning rule $A$ is* $\rho$-replicable *if*

$$\forall \text{ population } \mathcal{D}, \forall m: \; \mathbb{P}_{\substack{S_1, S_2 \sim \mathcal{D}^m \\ r \sim \mathcal{R}}}[A(S_1; r) = A(S_2; r)] \geq \rho,$$

*where $r$ represents the random coins of $A$.*

**Remark 3.9.** *Note that both in [BGH$^+$23] and in [ILPS22] the definition of $\rho$-replicability is slightly different. In their definition, they treat the parameter $\rho$ as the failure probability, i.e., $A$ is a $\rho$-replicable learning rule by their definition if the probability that $A(S_1; r) = A(S_2; r)$ is at least $1 - \rho$.*

There exists an alternative 2-parameter definition of replicability introduced in [ILPS22].

**Definition 3.10** ($(\eta, \nu)$-Replicability; [BGH$^+$23, ILPS22]). *Let $\eta, \nu \in \mathbb{R}_{>0}$ and let $\mathcal{R}$ be a distribution over random strings. Coin tosses $r$ are* $\eta$-good *for a learning rule $A$ with respect to a population distribution $\mathcal{D}$ if there exists a canonical output $h_r$ such that for every $m$, $\mathbb{P}_{S \sim \mathcal{D}^m}[A(S; r) = h_r] \geq \eta$. A learning rule $A$ is* $(\eta, \nu)$-replicable *if*

$$\forall \text{ population } \mathcal{D}: \; \mathbb{P}_{r \sim \mathcal{R}}[r \text{ is } \eta\text{-good}] \geq \nu.$$

### 3.3.5 Global Stability

**Definition 3.11** (Global Stability; [BLM20]). *Let $\eta > 0$ be a global stability parameter. A learning rule $A$ is* $(m, \eta)$-globally stable *with respect to a population distribution $\mathcal{D}$ if there exists a canonical output $h$ such that $\mathbb{P}[A(S) = h] \geq \eta$, where the probability is over $S \sim \mathcal{D}^m$ as well as the internal randomness of $A$.*

### 3.3.6 MI-Stability

**Definition 3.12** (Mutual Information Stability; [XR17, BMN$^+$18]). *A learning rule $A$ is* MI-stable *if there exists $f : \mathbb{N} \to \mathbb{N}$ with $f = o(m)$ such that*

$$\forall \text{ population } \mathcal{D} \; \forall m \in \mathbb{N}: I(A(S), S) \leq f(m),$$

*where $S \sim \mathcal{D}^m$.*

### 3.3.7  TV-Stability

**Definition 3.13** (TV-Stability; Appendix A.3.1 in [KKMV23])**.** *Let $A$ be a learning rule, and let $f : \mathbb{N} \to \mathbb{N}$ satisfy $f(m) = o(1)$.*

1. *$A$ is distribution-independent TV-stable if*

$$\exists \, prior \, \mathcal{P} \, \forall \, population \, \mathcal{D} \, \forall m \in \mathbb{N}: \; \mathbb{E}_{S \sim \mathcal{D}^m}[\mathsf{TV}(A(S), \mathcal{P})] \leq f(m).$$

2. *$A$ is distribution-dependent TV-stable if*

$$\forall \, population \, \mathcal{D} \, \exists \, prior \, \mathcal{P}_{\mathcal{D}} \, \forall m \in \mathbb{N}: \; \mathbb{E}_{S \sim \mathcal{D}^m}[\mathsf{TV}(A(S), \mathcal{P}_{\mathcal{D}})] \leq f(m).$$

### 3.3.8  Max Information

**Definition 3.14.** *Let $A$ be a learning rule, and let $\varepsilon, \delta \in \mathbb{R}_{\geq 0}$. $A$ has $(\varepsilon, \delta)$-max-information with respect to product distributions if for every event $\mathcal{O}$ we have*

$$\mathbb{P}[(A(S), S) \in \mathcal{O}] \leq e^{\varepsilon}\mathbb{P}[(A(S), S') \in \mathcal{O}] + \delta$$

*where are $S, S'$ are independent samples drown i.i.d from a population distribution $\mathcal{D}$.*

## Acknowledgements

SM is a Robert J. Shillman Fellow; he acknowledges support by ISF grant 1225/20, by BSF grant 2018385, by an Azrieli Faculty Fellowship, by Israel PBC-VATAT, by the Technion Center for Machine Learning and Intelligent Systems (MLIS), and by the the European Union (ERC, GENERALIZATION, 101039692). HS acknowledges support by ISF grant 1225/20, and by the the European Union (ERC, GENERALIZATION, 101039692). Views and opinions expressed are however those of the author(s) only and do not necessarily reflect those of the European Union or the European Research Council Executive Agency. Neither the European Union nor the granting authority can be held responsible for them. JS was supported by DARPA (Defense Advanced Research Projects Agency) contract #HR001120C0015 and the Simons Collaboration on The Theory of Algorithmic Fairness.

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

# A Proof for Theorem 2.2 (Stability Boosting)

## A.1 Information Theoretic Preliminaries

**Lemma A.1** (Monotonicity of Rényi divergence; Theorem 3 in [vEH14]). *Let $0 \leq \alpha < \beta \leq \infty$. Then $\mathsf{D}_\alpha(\mathcal{P} \parallel \mathcal{Q}) \leq \mathsf{D}_\beta(\mathcal{P} \parallel \mathcal{Q})$. Furthermore, the inequality is an equality if and only if $\mathcal{P}$ equals the conditional $\mathcal{Q}(\cdot \mid A)$ for some event $A$.*

**Lemma A.2** (Data Processing Inequality; Theorem 9 and Eq. 13 in [vEH14]). *Let $\alpha \in [0, \infty]$. Let $X$ and $Y$ be random variables, and let $F_{Y|X}$ be the law of $Y$ given $X$. Let $\mathcal{P}_Y, \mathcal{Q}_Y$ be the distributions of $Y$ when $X$ is sampled from $\mathcal{P}_X, \mathcal{Q}_X$, respectively. Then*

$$\mathsf{D}_\alpha(\mathcal{P}_Y \parallel \mathcal{Q}_Y) \leq \mathsf{D}_\alpha(\mathcal{P}_X \parallel \mathcal{Q}_X).$$

One interpretation of this is that processing an observation makes it more difficult to determine whether it came from $\mathcal{P}_X$ or $\mathcal{Q}_X$.

**Definition A.3** (Conditional KL-divergence; Definition 2.12 in [PW]). *Given joint distributions $\mathcal{P}(x,y), \mathcal{Q}(x,y)$, the* KL*-divergence of the marginals $\mathcal{P}(y|x), \mathcal{Q}(y|x)$ is*

$$\mathsf{KL}(\mathcal{P}(y|x) \parallel \mathcal{Q}(y|x)) = \sum_x \mathcal{P}(x) \sum_y \mathcal{P}(y|x) \log \frac{\mathcal{P}(y|x)}{\mathcal{Q}(y|x)}.$$

**Lemma A.4** (Chain Rule for KL-divergence; Theorem 2.13 in [PW]). *Let $\mathcal{P}(x,y), \mathcal{Q}(x,y)$ be joint distributions. Then,*

$$\mathsf{KL}(\mathcal{P}(x,y) \parallel \mathcal{Q}(x,y)) = \mathsf{KL}(\mathcal{P}(x) \parallel \mathcal{Q}(x)) + \mathsf{KL}(\mathcal{P}(y|x) \parallel \mathcal{Q}(y|x)).$$

**Lemma A.5** (Conditioning increases KL-divergence; Theorem 2.14(e) in [PW]). *For a distribution $\mathcal{P}_X$ and conditional distributions $\mathcal{P}_{Y|X}, \mathcal{Q}_{Y|X}$, let $\mathcal{P}_Y = \mathcal{P}_{Y|X} \circ \mathcal{P}_X$ and $\mathcal{Q}_Y = \mathcal{Q}_{Y|X} \circ \mathcal{P}_X$, where '$\circ$' denotes composition (see Section 2.4 in [PW]) Then*

$$\mathsf{KL}(\mathcal{P}_Y \parallel \mathcal{Q}_Y) \leq \mathsf{KL}\left(\mathcal{P}_{Y|X} \parallel \mathcal{Q}_{Y|X} \mid \mathcal{P}_X\right),$$

*with equality if and only if $\mathsf{KL}\left(\mathcal{P}_{X|Y} \parallel \mathcal{Q}_{X|Y} \mid P_Y\right) = 0$.*

## A.2 Online Learning Preliminaries

Following is some basic background on the topic of online learning with expert advice. This will be useful in the proof of Theorem 2.2.

Let $Z = \{z_1, \ldots, z_m\}$ be a set of experts and $I$ be a set of instances. For any instance $i \in I$ and expert $z \in Z$, following the advice of expert $z$ on instance $i$ provides utility $u(z, i) \in \{0, 1\}$.

The online learning setting is a perfect-information, zero-sum game between two players, a *learner* and an *adversary*. In each round $t = 1, \ldots, T$:

1. The learner chooses a distribution $w_t \in \Delta(Z)$ over the set of experts.

2. The adversary chooses an instance $i_t \in I$.

3. The learner gains utility $u_t = \mathbb{E}_{z \sim w_t}[u(z, i_t)]$.

The *total utility* of a learner strategy $\mathcal{L}$ for the sequence of instances chosen by the adversary is

$$U(\mathcal{L}, T) = \sum_{t=1}^T u_t.$$

The *regret* of the learner is the difference between the utility of the best expert and the learner's utility. Namely, for each $z \in Z$, let

$$U(z, T) = \sum_{t=1}^T u(z, i_t)$$

be the utility the learner would have gained had they chosen $w_t(z) = \mathbb{1}(z = z_j)$ for all $t \in [T]$. Then the regret is

$$\mathsf{Regret}(\mathcal{L}, T) = \max_{z \in Z} U(z, T) - U(\mathcal{L}, T).$$

There are several well-studied algorithms for online learing using expert advice that guarantee regret sublinear in $T$ for every possible sequence of $T$ instances. A classic example is the *Multiplicative Weights* algorithm (e.g., Section 21.2 in [SB14]), which enjoys the following guarantee.

**Theorem A.6** (Online Regret Bound). *In the setting of online learning with expert advice, there exists a learner strategy $\mathcal{L}$ such that for any sequence of $T$ instances selected by the adversary,*

$$\mathsf{Regret}(\mathcal{L}, T) \leq \sqrt{2T \log(m)},$$

*where $m$ is the number of experts.*

### A.3 Proof

**Theorem** (Theorem 2.2, Restatement). *Let $\mathcal{X}$ be a set, let $\mathcal{H} \subseteq \{0,1\}^{\mathcal{X}}$ be a hypothesis class, and let $A$ be a learning rule. Assume there exists $k \in \mathbb{N}$ and $\gamma > 0$ such that*

$$\forall \mathcal{D} \in \mathsf{Realizable}(\mathcal{H}) : \ \mathbb{E}_{S \sim \mathcal{D}^k}[\mathrm{L}_{\mathcal{D}}(A(S))] \leq \frac{1}{2} - \gamma, \tag{6}$$

*and there exists $\mathcal{P} \in \Delta\big(\{0,1\}^{\mathcal{X}}\big)$ and $b \geq 0$ such that*

$$\forall \mathcal{D} \in \mathsf{Realizable}(\mathcal{H}) : \ \mathbb{E}_{S \sim \mathcal{D}^k}[\mathsf{KL}(A(S) \,\|\, \mathcal{P})] \leq b. \tag{7}$$

*Then, there exists an interpolating learning rule $A^{\star}$ that PAC learns $\mathcal{H}$ with logarithmic $\mathsf{KL}$-stability. More explicitly, there exists a prior distribution $\mathcal{P}^{\star} \in \Delta\big(\{0,1\}^{\mathcal{X}}\big)$ and function $b^{\star}$ and $\varepsilon^{\star}$ that depend on $\gamma$ and $b$ such that*

$$\forall \mathcal{D} \in \mathsf{Realizable}(\mathcal{H}) \ \forall m \in \mathbb{N} :$$

$$\mathbb{P}_{S \sim \mathcal{D}^m}[\mathsf{KL}(A^{\star}(S) \,\|\, \mathcal{P}^{\star}) \leq b^{\star}(m) = O(\log(m))] = 1, \tag{8}$$

$$and$$

$$\mathbb{E}_{S \sim \mathcal{D}^m}[\mathrm{L}_{\mathcal{D}}(A^{\star}(S))] \leq \varepsilon^{\star}(m) = O\left(\sqrt{\frac{\log(m)}{m}}\right). \tag{9}$$

---

**Assumptions:**
- $\gamma, b > 0$; $m, k \in \mathbb{N}$.
- $S = \{(x_1, y_1), \ldots, (x_m, y_m)\}$ is an $\mathcal{H}$-realizable sample.
- $\mathcal{O}_S$ is the online learning algorithm of Appendix A.2, using expert set $S$.
- $T = \lceil 8 \log(m)/\gamma^2 \rceil + 1$.
- $A$ satisfies Eqs. (6) and (7) (with respect to $k, b, \gamma$).

$A^{\star}(S)$:
   **for** $t = 1, \ldots, T$:
      $w_t \leftarrow$ expert distribution chosen by $\mathcal{O}_S$ for round $t$
      **do**:
         sample $S_t \leftarrow (w_t)^k$
      **while** $\mathsf{KL}(A(S_t) \,\|\, \mathcal{P}) \geq 2b/\gamma$         ▷ See Remark A.7
      $f_t \leftarrow A(S_t)$
      $\mathcal{O}_S$ receives instance $f_t$ and gains utility $\mathbb{E}_{(x,y) \sim w_t}[\mathbb{1}(f_t(x) \neq y)]$
   **return** $\mathsf{Maj}(f_1, \ldots, f_T)$

Algorithm 1: The stability-boosted learning rule $A^{\star}$, which uses $A$ as a subroutine.

*Proof of Theorem 2.2.* Let $\mathcal{D} \in \mathsf{Realizable}(\mathcal{H})$ and $m \in \mathbb{N}$. Learning rule $A^{\star}$ operates as follows. Given a sample $S = \{(x_1, y_1), \ldots, (x_m, y_m)\}$, $A^{\star}$ simulates an online learning game, in which $S$ is the set of 'experts', $\mathcal{F} = \{0,1\}^{\mathcal{X}}$ is the set of 'instances', and the learner's utility for playing expert $(x, y)$ on instance $f \in \mathcal{F}$ is $\mathbb{1}(f(x) \neq y)$. Namely, in this game the learner is attempting to select an $(x, y)$ pair that disagrees with the instance $f$.

In this simulation, the learner executes an instance of the online learning algorithm of Appendix A.2 with expert set $S$. Denote this instance $\mathcal{O}_S$.

The adversary's strategy is as follows. Recall that at each round $t$, $\mathcal{O}_S$ chooses a distribution $w_t$ over $S$. Note that if $S$ is realizable then so is $w_t$. At each round $t$, the adversary selects an instance $f \in \mathcal{F}$ by executing $A$ on a training set sampled from $w_t$, as in Algorithm 1.

We prove the following:

1. $A^\star$ interpolates, namely $\mathbb{P}[\mathrm{L}_S(A^\star(S)) = 0] = 1$.

2. $A^\star$ has logarithmic KL-stability, as in Eq. (8).

3. $A^\star$ PAC learns $\mathcal{H}$ as in Eq. (9).

For Item 1, assume for contradiction that $A^\star$ does not interpolate. Seeing as $A^\star$ outputs $\mathsf{Maj}(f_1, \ldots, f_T)$, there exists an index $i \in [m]$ such that

$$\frac{T}{2} \leq \sum_{t=1}^{T} \mathbb{1}(f_t(x_i) \neq y_i) = U(i, T), \tag{10}$$

where $U(i, T)$ is the utility of always playing expert $i$ throughout the game.

Let $\mathcal{E}_t$ denote the event that $S_t$ was resampled (i.e., there were multiple iterations of the do-while loop in round $t$). Eq. (7) and Markov's inequality imply

$$\mathbb{P}[\mathcal{E}_t] = \mathbb{P}[\mathsf{KL}(A(S_t) \| \mathcal{P}) \geq 2b/\gamma] \leq \gamma/2. \tag{11}$$

The utility of $\mathcal{O}_S$ at time $t$ is

$$u_t^{\mathcal{O}_S} = \mathop{\mathbb{E}}_{\substack{S_t \sim (w_t)^k \\ f_t \sim A(S_t) \\ (x,y) \sim w_t}} [\mathbb{1}(f_t(x) \neq y)]$$

$$\leq \mathop{\mathbb{E}}_{S_t \sim (w_t)^k} [\mathrm{L}_{w_t}(A(S_t)) \mid \neg \mathcal{E}_t] + \mathbb{P}[\mathcal{E}_t] \leq \left(\frac{1}{2} - \gamma\right) + \frac{\gamma}{2},$$

where the last inequality follows from Eqs. (6) and (11). Hence, the utility of $\mathcal{O}_S$ throughout the game is

$$U(\mathcal{O}_S, T) = \sum_{t=1}^{T} u_t^{\mathcal{O}_S} \leq \left(\frac{1}{2} - \frac{\gamma}{2}\right) \cdot T. \tag{12}$$

Combining Eqs. (10) and (12) and Theorem A.6 yields

$$\frac{\gamma}{2} \cdot T \leq U(i, T) - U(\mathcal{O}_S, T) \leq \mathsf{Regret}(\mathcal{O}_S, T) \leq \sqrt{2T \log(m)},$$

which is a contradiction for our choice of $T$. This establishes Item 1.

For Item 2, for every $\ell \in \mathbb{N}$ let $\mathcal{P}_\ell^\star \in \Delta(\{0,1\}^{\mathcal{X}})$ be the distribution of $\mathsf{Maj}(g_1, \ldots, g_\ell)$, where $(g_1, \ldots, g_\ell) \sim \mathcal{P}^\ell$. Let $\mathcal{P}^\star = \frac{1}{z} \sum_{\ell=1}^{\infty} \mathcal{P}_\ell^\star / \ell^2$ where $z = \sum_{\ell=1}^{\infty} 1/\ell^2 = \pi^2/6$ is a normalization factor.

For any $S \in (\mathcal{X} \times \{0,1\})^m$,

$$\mathsf{KL}(A^\star(S) \| \mathcal{P}_T^\star) = \mathsf{KL}(\mathsf{Maj}(f_1, \ldots, f_T) \| \mathsf{Maj}(g_1, \ldots, g_T))$$

$$\leq \mathsf{KL}((f_1, \ldots, f_T) \| (g_1, \ldots, g_T)) \qquad \text{(By Lemma A.2)}$$

$$= \sum_{t=1}^{T} \mathsf{KL}((f_t|f_{<t}) \| (g_t|g_{<t})) \qquad \text{(By Lemma A.4)}$$

$$= \sum_{t=1}^{T} \mathsf{KL}((f_t|f_{<t}) \| g_t). \qquad \text{($g_i$'s are independent)}$$

$$= \sum_{t=1}^{T} \mathsf{KL}(A(S_t) \| \mathcal{P}) \leq T \cdot 2b/\gamma = O(\log(m)), \tag{13}$$

where the last inequality is due to the do-while loop in Algorithm 1. For any $S \in (\mathcal{X} \times \{0,1\})^m$,

$$
\begin{aligned}
\mathsf{KL}(A^\star(S) \,\|\, \mathcal{P}^\star) &= \mathbb{E}_{h \sim P_{A^\star(S)}} \left[ \log \left( \frac{P_{A^\star(S)}(h)}{\mathcal{P}^\star(h)} \right) \right] \\
&\leq \mathbb{E}_{h \sim P_{A^\star(S)}} \left[ \log \left( \frac{P_{A^\star(S)}(h)}{\mathcal{P}_T^\star(h)/(zT^2)} \right) \right] \\
&= \mathsf{KL}(A^\star(S) \,\|\, \mathcal{P}_T^\star) + O(\log(T)) = O(\log(m)). \qquad \text{(By Eq. (13))}
\end{aligned}
$$

This establishes Item 2.

Item 3 follows by plugging $\beta = \frac{1}{m}$ and Items 1 and 2 in the PAC-Bayes theorem (Theorem 3.2), yielding

$$
\mathbb{P}_{S \sim \mathcal{D}^m} \left[ \mathsf{L}_\mathcal{D}(A^\star(S)) \leq O\left( \sqrt{\frac{\log(m)}{m}} \right) \right] \geq 1 - \frac{1}{m}.
$$

This implies Item 3 because the 0-1 loss is at most 1. $\qquad \square$

**Remark A.7.** *Our definition of the learning rule $A^\star$ depends on $A$ and $\mathcal{P}$. The mapping $S_t \mapsto \mathsf{KL}(A(S_t) \,\|\, \mathcal{P})$ is well-defined, so $A^\star$ is a well-defined learning rule.[2]*

# B    Proof of Theorem 2.1 (DI Equivalences)

In this section, we prove Theorem 2.1.

**Theorem** (Theorem 2.1, Restatement). *Let $\mathcal{H}$ be a hypothesis class. The following is equivalent.*

1. *There exists a learning rule that PAC learns $\mathcal{H}$ and satisfied pure differential privacy (Definition 3.5).*

2. *$\mathcal{H}$ has finite fractional clique dimension.*

3. *For every $\alpha \in [1, \infty]$, there exists a learning rule that PAC learns $\mathcal{H}$ and satisfied distribution-independent $\mathsf{D}_\alpha$-stability (Definition 3.6).*

4. *For every $\alpha \in [1, \infty]$, there exists a distribution-independent $\mathsf{D}_\alpha$-stable PAC learner $A$ for $\mathcal{H}$, that satisfies the following:*

    (i) *$A$ is interpolating almost surely. Namely, for every $\mathcal{H}$-realizable distribution $\mathcal{D}$, $\mathbb{P}_{S \sim \mathcal{D}^m}[\mathsf{L}_S(A(S)) = 0] = 1$.*

    (ii) *$A$ admits a divergence bound of $f(m) = O(\log m)$, with confidence $\beta(m) \equiv 0$. I.e., for every $\mathcal{H}$-realizable distribution $\mathcal{D}$, $\mathsf{D}_\alpha(A(S) \,\|\, \mathcal{P}) \leq O(\log m)$ with probability 1, where $S \sim \mathcal{D}^m$ and $\mathcal{P}$ is a prior distribution independent of $\mathcal{D}$.*

    (iii) *For every $\mathcal{H}$-realizable distribution $\mathcal{D}$, the expected population loss of $A$ with respect to $\mathcal{D}$ satisfies $\mathbb{E}_{S \sim \mathcal{D}^m}[\mathsf{L}_\mathcal{D}(A(S))] \leq O\left( \sqrt{m^{-1} \log m} \right)$.*

*In particular, plugging $\alpha = 1$ in Item (ii) implies $\mathsf{KL}$-stability with divergence bound of $f(m) = O(\log m)$ and confidence $\beta(m) \equiv 0$. Plugging $\alpha = \infty$ implies distribution-independent one-way $\varepsilon$-pure perfect generalization, with $\varepsilon(m) \leq O(\log m)$ and confidence $\beta(m) \equiv 0$.*

The next subsections contain Theorem B.1, which is a useful result from [AMSY23], followed by the statements and proofs of Lemmas B.2 and B.4, which rely on Theorem B.1 and our boosting result (Theorem 2.2). The proof of Theorem 2.1 is a consequence of these results, as follows.

---

[2]We remark that if $A$ is a randomized Turing machine, then $\mathsf{KL}(A(S_t) \,\|\, \mathcal{P})$ can be estimated to arbitrary precision by a Turing machine with oracle access to the function $\mathcal{P}$. Namely, consider a Turing machine that can query an oracle for the value of $\mathcal{P}(h)$ up to precision $2^{-q}$ for any $h$ and $q \in \mathbb{N}$ of its choosing. To see that such a machine can estimate $\mathsf{KL}(A(S_t) \,\|\, \mathcal{P})$, observe that if $A$ uses some finite number of random coins, then $A(S_t)$ has a finite support, and so computing $\mathsf{KL}(A(S_t) \,\|\, \mathcal{P})$ involves querying $\mathcal{P}$ at a finite number of locations. Moreover, if $A$ uses a number $R$ of random coins, which is itself a random variable that may be unbounded but satisfies $\mathbb{E}[R] < \infty$, then by Markov's inequality there exists an explicit algorithm $A'$ that uses at most $\mathbb{E}[R]/\alpha$ random coins, such that $\mathsf{TV}(A(S_t), A'(S_t)) < \alpha$. Hence, $\mathsf{KL}(A'(S_t) \,\|\, \mathcal{P})$ can be estimated to arbitrary precision as before. Taking small enough values of $\alpha$ yields a modified version of $A^\star$ that can be shown to satisfy the requirements of Theorem 2.2.

*Proof of Theorem 2.1.* The proof follows from:

$$\text{Item 1} \overset{\text{Theorem B.1}}{\Longleftrightarrow} \text{Item 2} \overset{\text{Lemma B.2}}{\Longrightarrow} \text{Item 4} \overset{(*)}{\Longrightarrow} \text{Item 3} \overset{\text{Lemma B.4}}{\Longrightarrow} \text{Item 2},$$

where $(*)$ is immediate. $\qquad\square$

## B.1 Characterization of Pure DP Learnability via the Fractional Clique Dimension

For every hypothesis class $\mathcal{H}$, they define a quantity $\omega_m^\star = \omega_m^\star(\mathcal{H})$, called the *fractional clique number* of $\mathcal{H}$. The definition of $\omega_m^\star$ involves an LP relaxation of clique numbers on a certain graph corresponding to $\mathcal{H}$, but for our purposes it will be more convenient to use the following alternative characterization (Eq. 6 and Theorem 2.8 in [AMSY23]):

$$\forall m \in \mathbb{N}: \ \frac{1}{\omega_m^\star} = \sup_{\mathcal{P}} \inf_{\mathcal{S}} \underset{\substack{S \sim \mathcal{S} \\ h \sim \mathcal{P}}}{\mathbb{P}}[\mathrm{L}_S(h) = 0], \tag{14}$$

where the supremum is taken over distributions over $\mathcal{H}$, and the infimum is taken over distributions over samples of size $m$ that are realizable by $\mathcal{H}$. In words, $1/\omega_m^\star$ is the value of a game in which player 1 selects a distribution of hypotheses over $\mathcal{H}$, player 2 selects a distribution over realizable samples of size $m$, and player 1 wins if and only if the hypothesis correctly labels all the points in the sample.

The fractional clique number characterizes pure DP learnability, as follows:

**Theorem B.1** (Restatement of Theorems 2.3 and 2.6 in [AMSY23])**.** *For any hypothesis class $\mathcal{H}$, exactly one of the following statements holds:*

1. *$\mathcal{H}$ is pure DP learnable (as in Definition 3.5), and there exists a polynomial $p$ such that $\omega_m^\star(\mathcal{H}) \leq p(m)$ for all $m \in \mathbb{N}$.*

2. *$\mathcal{H}$ is not pure DP learnable, and $\omega_m^\star(\mathcal{H}) = 2^m$ for all $m \in \mathbb{N}$.*

The *fractional clique dimension* of $\mathcal{H}$ is defined by $\mathsf{CD}^\star(\mathcal{H}) = \sup\{m \in \mathbb{N}: \omega_m^\star(\mathcal{H}) = 2^m\}$. So in other words, Theorem B.1 states that $\mathcal{H}$ is pure DP learnable if and only if $\mathsf{CD}^\star(\mathcal{H})$ is finite.

## B.2 Finite Fractional Clique Dimension $\implies$ DI Rényi-Stability

**Lemma B.2.** *In the context of Theorem 2.1: Item 2 $\implies$ Item 4.*

*Proof of Lemma B.2.* Given that $\mathcal{H}$ is DP learnable, we define a learning rule $A$ and a prior $\mathcal{P}$, and show that $A$ PAC learns $\mathcal{H}$ subject to distribution-independent KL-stability with respect to $\mathcal{P}$.

By Theorem B.1 there exists a polynomial $p$ such that $\omega_m^\star(\mathcal{H}) \leq p(m)$ for all $m \in \mathbb{N}$. By Eq. (14), for every $m \in \mathbb{N}$, there exists a prior $\mathcal{P}_m \in \Delta(\{0,1\}^{\mathcal{X}})$ such that for any $\mathcal{H}$-realizable sample $S \in (\mathcal{X} \times \{0,1\})^m$,

$$\underset{h \sim \mathcal{P}_m}{\mathbb{P}}[\mathrm{L}_S(h) = 0] \geq \frac{1}{\omega_m^\star} \geq \frac{1}{p(m)}.$$

Let

$$\mathcal{P} = \frac{1}{z} \sum_{m=1}^{\infty} \frac{\mathcal{P}_m}{m^2}$$

be a mixture, where $z = \sum_{m=1}^{\infty} 1/m^2 = \pi^2/6$ is a normalization factor. $\mathcal{P}$ is a valid distribution over $\{0,1\}^{\mathcal{X}}$.

For every $m \in \mathbb{N}$ and for any $\mathcal{H}$-realizable sample $S \in (\mathcal{X} \times \{0,1\})^m$,

$$\underset{h \sim \mathcal{P}}{\mathbb{P}}[\mathrm{L}_S(h) = 0] \geq \frac{1}{zm^2} \cdot \underset{h \sim \mathcal{P}_m}{\mathbb{P}}[\mathrm{L}_S(h) = 0] \geq \frac{1}{zm^2 p(m)} = \frac{1}{q(m)}, \tag{15}$$

where $q(m) = zm^2 p(m)$.

For any sample $S$, let $C_S = \{h \in \{0,1\}^{\mathcal{X}}: \mathrm{L}_S(h) = 0\}$ be the set of hypotheses consistent with $S$. Let $A$ be a randomized learning rule given by $S \mapsto \mathcal{Q}_S \in \Delta(\{0,1\}^{\mathcal{X}})$ such that $\mathcal{Q}_S(h) = \mathcal{P}(h \mid C_S)$ if $h \in C_S$, and $\mathcal{Q}_S(h) = 0$ otherwise. $A$ can be written explicitly as a rejection sampling algorithm:

```
A(S):
    do:
        sample h ← 𝒫
    while L_S(h) > 0
    return h
```

Algorithm $A$ terminates with probability 1, because for any realizable sample $S$ of size $m \in \mathbb{N}$ and any $t \in \mathbb{N}$,

$$\mathbb{P}[A \text{ did not terminate after } t \text{ iterations}] = (\mathbb{P}_{h \sim \mathcal{P}}[L_S(h) > 0])^t \leq \left(1 - \frac{1}{q(m)}\right)^t \xrightarrow{t \to \infty} 0,$$

where the inequality follows by Eq. (15).

To complete the proof, we show that $A$ satisfies *(i)*, *(ii)* and *(iii)* in Item 4.

Item *(i)* is immediate from the construction of $A$. For Item *(ii)*, let $m \in \mathbb{N}$. For any sample $S$ of size $m$ and hypothesis $h \in C_S$,

$$\mathcal{Q}_S(h) = \mathcal{P}(h \mid C_S) = \frac{\mathcal{P}(\{h\} \cap C_S)}{\mathcal{P}(C_S)} \leq q(m) \cdot \mathcal{P}(h), \tag{16}$$

where the inequality follows from Eq. (15). Hence,

$$\begin{aligned}
\mathsf{D}_\infty(\mathcal{Q}_S \,\|\, \mathcal{P}) &= \log \left( \operatorname*{ess\,sup}_{\mathcal{Q}_S} \frac{\mathcal{Q}_S(h)}{\mathcal{P}(h)} \right) \\
&\leq \log \left( \operatorname*{ess\,sup}_{\mathcal{Q}_S} \frac{q(m) \cdot \mathcal{P}(h)}{\mathcal{P}(h)} \right) \qquad \text{(from Eq. (16) and } \mathcal{Q}_S(C_S) = 1) \\
&\leq \log(q(m)) = O(\log(m)).
\end{aligned}$$

Item *(ii)* follows from monotonicity of $\mathsf{D}_\alpha$ with respect to $\alpha$ (Lemma A.1). In particular, $\mathsf{KL}(\mathcal{Q}_S \,\|\, \mathcal{P}) = O(\log(m))$.

Item *(iii)* follows from the PAC-Bayes theorem (Theorem 3.2). Indeed, take $\beta = \frac{1}{m}$ and note that $L_S(\mathcal{Q}_S) = 0$ for all realizable $S$. Then for any $\mathcal{H}$-realizable distribution $\mathcal{D}$,

$$\mathbb{P}_{S \sim \mathcal{D}^m} \left[ L_\mathcal{D}(A(S)) \leq \sqrt{\frac{\mathsf{KL}(\mathcal{Q}_S \,\|\, \mathcal{P}) + 2\ln m}{2(m-1)}} \right] \geq 1 - \frac{1}{m}.$$

This implies that for any $\mathcal{H}$-realizable distribution $\mathcal{D}$,

$$\mathbb{E}_{S \sim \mathcal{D}^m}[L_\mathcal{D}(A(S))] \leq \frac{1}{m} + \sqrt{\frac{\mathsf{KL}(\mathcal{Q}_S \,\|\, \mathcal{P}) + 2\ln m}{2(m-1)}} = O\left(\sqrt{\frac{\log m}{m}}\right),$$

as desired. $\qquad \square$

**Remark B.3.** *The 'furthermore' section of Lemma A.1 implies that in the foregoing proof,* $\mathsf{D}_\alpha(\mathcal{Q}_S \,\|\, \mathcal{P}) = \mathsf{D}_\beta(\mathcal{Q}_S \,\|\, \mathcal{P})$ *for any* $\alpha, \beta \in [0, \infty]$.

### B.3 DI Rényi-Stability $\implies$ Finite Fractional Clique Dimension

**Lemma B.4.** *In the context of Theorem 2.1: Item 3 $\implies$ Item 2.*

*Proof of Lemma B.4.* By Theorem B.1 and Eq. (14) it suffices to show that there exist $m \in \mathbb{N}$ and a prior $\mathcal{P}$ such that for every $\mathcal{H}$-realizable sample $S \in (\mathcal{X} \times \{0,1\})^m$,

$$\mathbb{P}_{h \sim \mathcal{P}}[L_S(h) = 0] > \frac{1}{2^m}. \tag{17}$$

By the assumption (Item 3) and Theorem 2.2, there exists an interpolating learning rule $A^\star$, a prior $\mathcal{P}^\star$, and a constant $C > 0$ such that for every $\mathcal{D} \in \mathsf{Realizable}(\mathcal{H})$, the equality

$$\mathbb{P}_{S \sim \mathcal{D}^m}[\mathsf{KL}(A^\star(S) \,\|\, \mathcal{P}^\star) \leq C \log(m)] = 1 \tag{18}$$

holds for all $m \in \mathbb{N}$ large enough. Fix such an $m$. We show that taking $\mathcal{P} = \mathcal{P}^\star$ satisfies Eq. (17) for this $m$.

Let $\mathcal{Q}$ denote the distribution of $A^\star(S')$ where $S' \sim (\mathrm{U}(S))^{m'} = P_{S'}$, $\mathrm{U}(S)$ is the uniform distribution over $S$, and $m' = m \ln(4m)$. The proof follows by noting that if $\mathsf{KL}(\mathcal{Q} \,\|\, \mathcal{P}^\star)$ is small then one can lower bounding the probability of an event according to $\mathcal{P}^\star$ by its probability according to $\mathcal{Q}$.

To see that the KL is indeed small, let $P_{A^\star(S'),S'}$ and $P_{H^\star,S'}$ be two joint distributions. The variable $S'$ has marginal $P_{S'}$ in both distributions, $A^\star(S') \sim \mathcal{Q}$ depends on $S'$, but $H^\star \sim \mathcal{P}^\star$ is independent of $S'$. Then,

$$
\begin{aligned}
\mathsf{KL}(\mathcal{Q} \,\|\, \mathcal{P}^\star) &= \mathsf{KL}\big(P_{A^\star(S')} \,\|\, P_{H^\star}\big) \\
&\leq \mathsf{KL}\Big(P_{A^\star(S')|S'} \,\|\, P_{H^\star|S'} \,\Big|\, P_{S'}\Big) && \text{(Lemma A.5)} \\
&= \mathsf{KL}\Big(P_{A^\star(S')|S'} \,\|\, P_{H^\star} \,\Big|\, P_{S'}\Big) && (H^\star \!\perp\! S') \\
&= \mathbb{E}_{S'}[\mathsf{KL}(A^\star(S') \,\|\, \mathcal{P}^\star)] && \text{(Definition of conditional KL)} \\
&\leq C \log(m). && \text{(By Eq. (18) and choice of } m) \quad (19)
\end{aligned}
$$

Taking $k = 2C \log(m)$,

$$
\mathbb{P}_{h \sim \mathcal{Q}}\left[\log\left(\frac{\mathcal{Q}(h)}{\mathcal{P}^\star(h)}\right) \geq k\right] \leq \frac{\mathsf{KL}(\mathcal{Q} \,\|\, \mathcal{P}^\star)}{k} \leq \frac{1}{2} \qquad (20)
$$

holds by Markov's inequality and the definition of the KL divergence. We are interested in the probability of the event $\mathcal{E} = \{h \in \{0,1\}^{\mathcal{X}} : \mathrm{L}_S(h) = 0\}$. Because $A^\star$ is interpolating,

$$
\mathcal{Q}(\mathcal{E}) \geq \mathbb{P}_{\substack{S' \sim (\mathrm{U}(S))^{m'} \\ h \sim A^\star(S')}}[S \subseteq S'] \geq 1 - m\left(1 - \frac{1}{m}\right)^{m'} \geq \frac{3}{4}. \qquad (21)
$$

Finally, we lower bound $\mathcal{P}^\star(\mathcal{E})$ as follows.

$$
\begin{aligned}
\mathcal{P}^\star(\mathcal{E}) &\geq \mathbb{P}_{h \sim \mathcal{P}^\star}\left[\mathcal{E} \,\wedge\, \log\left(\frac{\mathcal{Q}(h)}{\mathcal{P}^\star(h)}\right) \leq k\right] \\
&= \mathbb{P}_{h \sim \mathcal{P}^\star}\left[\mathcal{E} \,\wedge\, \mathcal{P}^\star(h) \geq 2^{-k} \cdot \mathcal{Q}(h)\right] \\
&\geq \mathbb{P}_{h \sim \mathcal{Q}}\left[\mathcal{E} \,\wedge\, \mathcal{P}^\star(h) \geq 2^{-k} \cdot \mathcal{Q}(h)\right] \cdot 2^{-k} \\
&= \mathbb{P}_{h \sim \mathcal{Q}}\left[\mathcal{E} \,\wedge\, \log\left(\frac{\mathcal{Q}(h)}{\mathcal{P}^\star(h)}\right) \leq k\right] \cdot 2^{-k}. \\
&\geq \left(\mathcal{Q}(\mathcal{E}) - \mathbb{P}_{h \sim \mathcal{Q}}\left[\log\left(\frac{\mathcal{Q}(h)}{\mathcal{P}^\star(h)}\right) \leq k\right]\right) \cdot 2^{-k}. && \text{(De Morgan's + union bound)} \\
&\geq \frac{1}{4} \cdot 2^{-k} = \frac{1}{4m^{2C}} = \frac{1}{\mathsf{poly}(m)}. && \text{(By Eqs. (20) and (21) and choice of } k)
\end{aligned}
$$

This establishes Eq. (17), as desired. $\qquad \square$

## C   Proof of Theorem 1.4 (DD Equivalences)

### C.1   Preliminaries

#### C.1.1   Littlestone Dimension

The Littlestone dimension is a combinatorial parameter which captures mistake and regret bounds in online learning [Lit87, BPS09].

**Definition C.1** (Mistake Tree). *A mistake tree is a binary decision tree whose nodes are labeled with instances from $\mathcal{X}$ and edges are labeled by $0$ or $1$ such that each internal node has one outgoing edge labeled $0$ and one outgoing edge labeled $1$. A root-to-leaf path in a mistake tree can be described as a sequence of labeled examples $(x_1, y_1), \ldots, (x_d, y_d)$. The point $x_i$ is the label of the $i$-th internal node in the path, and $y_i$ is the label of its outgoing edge to the next node in the path.*

**Definition C.2** (Shattering). *Let $\mathcal{H}$ be a hypothesis class and let $T$ be a mistake tree. $\mathcal{H}$ shatters $T$ if every root-to-leaf path in $T$ is realizable by $\mathcal{H}$.*

**Definition C.3** (Littlestone Dimension). *Let $\mathcal{H}$ be a hypothesis class. The Littlestone dimension of $\mathcal{H}$, denoted $\mathsf{LD}(\mathcal{H})$, is the largest number $d$ such that there exists a complete mistake tree of depth $d$ shattered by $\mathcal{H}$. If $\mathcal{H}$ shatters arbitrarily deep mistake trees then $\mathsf{LD}(\mathcal{H}) = \infty$.*

### C.1.2 Clique Dimension

**Definition C.4** (Clique; [AMSY23]). *Let $\mathcal{H}$ be a hypothesis class and let $m \in \mathbb{N}$. A clique in $\mathcal{H}$ of order $m$ is a family $\mathcal{S}$ of realizable samples of size $m$ such that (i) $|\mathcal{S}| = 2^m$; (ii) every two distinct samples $S', S'' \in \mathcal{S}$ contradicts, i.e., there exists a common example $x \in \mathcal{X}$ such that $(x, 0) \in S'$ and $(x, 1) \in S''$.*

**Definition C.5** (Clique Dimension; [AMSY23]). *Let $\mathcal{H}$ be a hypothesis. The clique dimension of $\mathcal{H}$, denoted $\mathsf{CD}(\mathcal{H})$, is the largest number $m$ such that $\mathcal{H}$ contains a clique of order $m$. If $\mathcal{H}$ contains cliques of arbitrary large order then we write $\mathsf{CD}(\mathcal{H}) = \infty$.*

## C.2 Global Stability $\implies$ Replicability

**Lemma C.6.** *Let $\mathcal{H}$ be a hypothesis class and let $A$ be a $(m, \eta)$-globally stable learner for $\mathcal{H}$. Then, $A$ is an $\eta$-replicable learner for $\mathcal{H}$.*

This follows immediately by noting that global stability is equivalent to 2-parameters replicability, which is qualitatively equivalent to 1-parameter replicability [ILPS22].

**Lemma C.7** ([ILPS22]). *For every $\rho, \eta, \nu \in [0, 1]$,*

1. *Every $\rho$-replicable algorithm is also $\left(\frac{\rho - \nu}{1 - \nu}, \nu\right)$-replicable.*

2. *Every $(\eta, \nu)$-replicable algorithm is also $(\eta + 2\nu - 2)$-replicable.*

*Proof of Lemma C.6.* By the assumption, there exists an hypothesis $h$ such that for every population $\mathcal{D}$, we have $\mathbb{P}_{R \sim \mathcal{R}}[\mathbb{P}_{S \sim \mathcal{D}^m}[A(S; r) = h] \geq \eta] = 1$. Hence $A$ is $(\eta, 1)$-replicable, and by Lemma C.7 it is also $\eta$-replicable. $\square$

## C.3 DD KL-Stability $\implies$ Finite Littlestone Dimension

**Lemma C.8.** *Let $\mathcal{H}$ be a hypothesis class that is distribution-dependent KL-stable. Then $\mathcal{H}$ has finite Littlestone dimension.*

This lemma is an immediate result of the relation between thresholds and the Littlestone dimension, and the fact that the class of thresholds on the natural numbers does not admit any learning rule that satisfies a non-vacuous PAC-Bayes bound [LM20]. The next lemma is a corollary of Theorem 2 in [LM20].

**Theorem C.9** (Corollary of Theorem 2 [LM20]). *Let $m \in \mathbb{N}$ and let $N \in \mathbb{N}$. Then, there exists $n \in \mathbb{N}$ large enough such that the following holds. For every learning rule $A$ of the class of thresholds over $[n]$, $\mathcal{H}_n = \{\mathbb{1}_{[x > k]} : [n] \to \{0, 1\} \mid k \in [n]\}$, there exists a realizable population distribution $\mathcal{D} = \mathcal{D}_A$ such that for any prior distribution $\mathcal{P}$,*

$$\mathbb{P}_{S \sim \mathcal{D}^m}\left[\mathsf{KL}(A(S) \parallel \mathcal{P}) > N \quad or, \quad \mathrm{L}_\mathcal{D}(A(S)) > \frac{1}{4}\right] \geq \frac{1}{16}$$

**Theorem C.10** (Littlestone dimension and thresholds [She90]). *Let $\mathcal{H}$ be a hypothesis class. Then,*

1. *If $\mathsf{LD}(\mathcal{H}) \geq d$ then $\mathcal{H}$ contains $\lfloor \log d \rfloor$ thresholds.*

2. *If $\mathcal{H}$ contains $d$ thresholds then $\mathsf{LD}(\mathcal{H}) \geq \lfloor \log d \rfloor$.*

*Proof of Lemma C.8.* If by contradiction the Littlestone dimension of $\mathcal{H}$ is unbounded, then by Theorem C.10, $\mathcal{H}$ contains a copy of $\mathcal{H}_n$, the class of thresholds over $[n]$, for arbitrary large $n$'s. Hence, by Theorem C.9 $\mathcal{H}$ does not admit a PAC learner that is KL-stable. $\square$

### C.4  MI-**Stability** $\implies$ **DD** KL-**Stability**

**Lemma C.11.** *Let $\mathcal{H}$ be a hypothesis class and let $A$ be a mutual information stable learner with information bound $f(m) = o(1)$. (I.e. for every population distribution $\mathcal{D}$, $I(A(S); S) \leq f(m)$ where $S \sim \mathcal{D}^m$.) Then, $A$ is a distribution-dependent KL-stable learner with KL bound $g(m) = \sqrt{f(m) \cdot m}$ and confidence $\beta(m) = \sqrt{f(m)/m}$.*

The following statement is an immediate corollary.

**Corollary C.12.** *Let $\mathcal{H}$ be a hypothesis class that is mutual information stable. Then $\mathcal{H}$ is distribution-dependent KL-stable.*

*Proof of Lemma C.11.* Let $\mathcal{D}$ be a population distribution. Define a prior distribution $\mathcal{P}_{\mathcal{D}} = \mathbb{E}_S[A(S)]$, i.e. $\mathcal{P}_{\mathcal{D}}(h) = \mathbb{P}_{S \sim \mathcal{D}^m}[A(S) = h]$. We will show that $A$ is KL stable with respect to the prior $\mathcal{P}_{\mathcal{D}}$. We use the identity $I(X; Y) = \mathsf{KL}(P_{X,Y}, P_X P_Y)$. Let $P_{A(S),S}$ be the joint distribution of the training sample $S$ and the hypothesis selected by $A$ when given $S$ as an input, and let $P_{A(S)}P_S$ be the product of the marginals. Note that $P_{A(S)}P_S$ is equal in distribution to $P_{A(S')}P_S$, where $S'$ is an independent copy of $S$. Hence,

$$
\begin{aligned}
I(A(S); S) &= \mathsf{KL}(P_{A(S),S}, P_{A(S)}P_S) \\
&= \mathsf{KL}(P_{A(S)|S}P_S, P_{A(S')}P_S), \\
&= \mathsf{KL}(P_S, P_S) + \mathbb{E}_{s \sim P_S}\big[\mathsf{KL}(P_{A(S)|S=s}, P_{A(S')|S=s})\big] \qquad \text{(Chain rule)} \\
&= \mathbb{E}_{s \sim P_S}\big[\mathsf{KL}(P_{A(S)|S=s}, P_{A(S')|S=s})\big] \\
&= \mathbb{E}_{s \sim P_S}\big[\mathsf{KL}(P_{A(S)|S=s}, P_{A(S')})\big].
\end{aligned}
$$

Note that $P_{A(S')}$ and the prior $\mathcal{P}_{\mathcal{D}}$ are identically distributed, and $P_{A(S)|S=s}$ is exactly the posterior produced by $A$ given the input sample $s$. By Markov's inequality,

$$
\begin{aligned}
\mathbb{P}_{S \sim D^m}\Big[\mathsf{KL}(A(S) \parallel P_{\mathcal{D}}) \geq \sqrt{m \cdot I(A(S); S)}\Big] &\leq \frac{I(A(S); S)}{\sqrt{mI(A(S); S)}} \\
&= \sqrt{\frac{I(A(S); S)}{m}}.
\end{aligned} \tag{22}
$$

Since $I(A(S); S) \leq f(m)$, by Eq. (22)

$$
\mathbb{P}_{S \sim D^m}\Big[\mathsf{KL}(A(S) \parallel P_{\mathcal{D}}) \geq \sqrt{f(m) \cdot m}\Big] \leq \sqrt{\frac{f(m)}{m}}.
$$

Note that since $f(m) = o(m)$, indeed $\sqrt{f(m)/m} \xrightarrow{m \to \infty} 0$ and $\sqrt{f(m) \cdot m} = o(m)$. $\qquad\square$

### C.5  **Finite Littlestone Dimension** $\implies$ MI-**Stability**

**Lemma C.13.** *Let $\mathcal{H}$ be a hypothesis class with finite Littlestone dimension. Then $\mathcal{H}$ admits an information stable learner.*

This lemma is a direct result of Theorem 2 in [PNG22].

**Definition C.14.** *The information complexity of a hypothesis class $\mathcal{H}$ is*

$$
\mathsf{IC}(\mathcal{H}) = \sup_{|S|} \inf_A \sup_{\mathcal{D}} I(A(S); S)
$$

*where the supremum is over all sample sizes $|S| \in \mathbb{N}$ and the infimum is over all learning rules that PAC learn $\mathcal{H}$.*

**Theorem C.15** (Theorem 2 [PNG22])**.** *Let $\mathcal{H}$ be a hypothesis class of with Littlestone dimension $d$. Then the information complexity of $\mathcal{H}$ is bounded by*

$$
\mathsf{IC}(\mathcal{H}) \leq 2^d + \log(d+1) + 3 + \frac{3}{e \ln 2}.
$$

*Proof of Lemma C.13.* Since finite information complexity implies that $\mathcal{H}$ admits an information stable learner, the proof follows from Theorem C.15 $\qquad\square$

