# OpenReview forum: "The Bayesian Stability Zoo"
_NeurIPS.cc/2023/Conference — NeurIPS 2023 poster_

### Official Review · Reviewer_xEWt · 2023-06-28

**Soundness:** 2 fair
**Presentation:** 1 poor
**Contribution:** 3 good
**Rating:** 5
**Confidence:** 3

**Summary:**

This paper aims to establish the equivalences among various definitions of distribution independent stability and different definitions of distribution dependent stability. Furthermore, the authors propose a stability-boosted interpolating learning rule that exhibits logarithmic expansion of KL-stability with respect to the sample size.

**Strengths:**

1. The authors establish the equivalence between different definitions of stability.
2. The stability boosting results presented in Section 2.2 are particularly intriguing and can be considered the primary novelty of this paper. These results demonstrate the effectiveness of the proposed stability-boosted interpolating learning rule.

**Weaknesses:**

1. The paper is not well-written and is lack of self-containment. For example, the abbreviation "PAC" is used throughout the main text without providing its full name when it is initially mentioned, despite it being a well-known concept in machine learning. Additionally, the structure of the paper lacks proper organization, as a significant portion is dedicated to introducing existing results and preliminaries, while the main contributions of Theorem 2.1 and Theorem 2.2 are given relatively little emphasis.

2.The main weakness of the paper lies in the overlap between the main contribution and existing results, leading to insufficient novelty. Many existing studies, such as "From Robustness to Privacy and Back" by Asi et al. and those mentioned in Theorem 1.3 of this paper, have already established the equivalence between distribution independent stability and differential privacy. Consequently, the primary contribution of Theorem 2.1 and Figure 1 appears to be a review of these existing results, which is claimed as the main contribution in this paper.

Theorem 2.2, from my perspective, is an interesting and significant novelty in this paper. However, its importance is not adequately emphasized in the main body. After reading the construction of $A^*$ in Theorem 2.2 in the appendix, I believe that this construction should be highlighted in the main body to showcase its novelty.

3. One noticeable limitation of this paper is the lack of empirical evaluation for the weak learning rule $A$ and the stability-boosted learning rule $A^*$ in Theorem 2.2. As a result, it is uncertain whether this construction holds practical significance without empirical evidence.

**Questions:**

According to the weakness mentioned, it would be beneficial for the authors to improve the English writing of the paper and place more emphasis on the main contribution, supported by experimental results.

**Limitations:**

Yes

---

> ### Author Rebuttal · Authors · 2023-08-09
>
>  **The paper is not well-written and is lack of self-containment. For example, the abbreviation "PAC" is used throughout the main text without providing its full name when it is initially mentioned, despite it being a well-known concept in machine learning.**
>
> PAC learnability is defined on line 193 of the paper. You are correct that we neglected to state the full unabbreviated name ("Probably Approximately Correct''). We believe most readers would be familiar with this abbreviation, but nonetheless we will add the full unabbreviated name in the final version.
>
> **The structure of the paper lacks proper organization, as a significant portion is dedicated to introducing existing results and preliminaries, while the main contributions of Theorem 2.1 and Theorem 2.2 are given relatively little emphasis.**
>
> We believe that this work has to include a fair amount of previous work and preliminaries, because it studies connections between many notions of stability that exist in the literature. We agree with this reviewer that it is important that the paper be self-contained (as they mentioned), and this requires providing sufficient detail on existing notions.
>
> We are also happy to add additional discussion about our main theorems, including their intuitive meaning and significance.
>
> **The main weakness of the paper lies in the overlap between the main contribution and existing results, leading to insufficient novelty. [...] Consequently, the primary contribution of Theorem 2.1 and Figure 1 appears to be a review of these existing results, which is claimed as the main contribution in this paper.**
>
> This is not accurate. Theorem 2.1 (which the reviewer mentions) is in fact an original contribution of this paper and is NOT a review of existing results. It involves a non-trivial proof, an relies on our stability boosting result (Theorem 2.2). These are meaningful original contributions.
>
> Regarding Theorem 1.3 and Figure 1, these are indeed summaries of existing results. However, we believe that these parts of the paper constitute a meaningful conceptual contribution. Namely, our observation that a large number of disparate results can neatly be organized based on our notions of distribution-dependent and distribution-independent notions of stability is a valuable contribution that could help researchers make sense of the stability landscape.
>
> **Many existing studies, such as "From Robustness to Privacy and Back" by Asi et al. and those mentioned in Theorem 1.3 of this paper, have already established the equivalence between distribution independent stability and differential privacy.**
>
> The paper from "Robustness to Privacy and Back'' studies connections between privacy and *robustness to adversarial perturbations of the input*. This form of robustness is technically different from the baysian stability notions considered in our paper.
>
> **Theorem 2.2, from my perspective, is an interesting and significant novelty in this paper. However, its importance is not adequately emphasized in the main body. After reading the construction of $A^\star$ in Theorem 2.2 in the appendix, I believe that this construction should be highlighted in the main body to showcase its novelty.**
>
> Thank you for this positive appreciation! We are happy to further highlight the value and novelty of our construction in the final version of the paper.
>
> **One noticeable limitation of this paper is the lack of empirical evaluation for the weak learning rule $A$ and the stability-boosted learning rule $A^\star$ in Theorem 2.2. As a result, it is uncertain whether this construction holds practical significance without empirical evidence.**
>
> Indeed, it would be valuable to perform an empirical evaluation of our work. Our work is purely theoretical/mathematical, and we leave experimental evaluations to future research.

---

> > ### Comment · Reviewer_xEWt · 2023-08-10
> > **Response to the rebuttal.**
> >
> > Thank you for your comprehensive response. I am open to revising my rating once I receive satisfactory answers to the following additional questions:
> >
> > 1. Could you kindly elaborate further on the formulation and development of the stability-boosted learning rule $A^*$? Given the absence of empirical validation concerning its effectiveness, I would recommend that the authors furnish more intricate insights into the tightness of this theoretical framework. Alternatively, if available, references to established works supporting this approach would greatly enhance its credibility.
> >
> > 2. In addition to the insights presented in Theorem 2.2, could you highlight any other innovative contributions that distinguish your work from the prior literature as listed in Theorem 1.3? (This question arises from my uncertainty regarding whether the primary contribution of formulating the stability-boosted rule holds sufficient promise, as indicated in Question 1. Should Question 1 receive a satisfactory response even in the absence of an answer to Question 2, I would be happy to revise my rating accordingly.)

---

> > > ### Author Response · Authors · 2023-08-14
> > > **That is Wonderful!**
> > >
> > > Thank you so much for your time and your careful consideration. We are thrilled to hear that you are open to revising your rating!
> > >
> > > Regarding your questions:

---

> > > > ### Author Response · Authors · 2023-08-14
> > > > **Answer to The First Question**
> > > >
> > > >
> > > > **Could you kindly elaborate further on the formulation and development of the stability-boosted learning rule $A^\star$? Given the absence of empirical validation concerning its effectiveness, I would recommend that the authors furnish more intricate insights into the tightness of this theoretical framework. Alternatively, if available, references to established works supporting this approach would greatly enhance its credibility.**
> > > >
> > > > Yes, we are happy to provide additional background and support from established works concerning our stability boosting result.
> > > >
> > > > **Background**
> > > >
> > > > Boosting has been a central topic of study in computational learning theory since its inception in the 1990s by Schapire [1] and Freund [2]. Interestingly, it was originally devised as a proof strategy to show a lower bound on the computational complexity of PAC learning. It is essentially a reduction from standard PAC learning to a (seemingly) easier problem of “weak learning”, similar to the many other reductions in computational complexity (e.g., between NP complete problems). However, it quickly became apparent that this reduction actually provides a very powerful and efficient way to perform PAC learning. The best-known boosting algorithm is AdaBoost [3], which has been extensively studied ([3] has >20k citations), and it is also widely used in practice as an excellent out-of-the-box ML algorithm that requires minimal tuning. Boosting also has rich connections with other topics such as game theory, online learning, and convex optimization (see [4], Ch. 5 in [5], Ch. 7 in [6]).
> > > >
> > > > **Our Result**
> > > >
> > > > In Theorem 2.2 of our paper, we present a novel boosting result. This results shows that given any learning algorithm $A$ that
> > > >  * Performs slightly better than random guessing (i.e., has expected loss at most $1/2-\gamma$ for some small positive $\gamma$), and
> > > >  * Is slightly KL-stable (i.e., the expected KL divergence from some fixed prior is not infinite),
> > > > We can construct a new learning algorithm $A^*$ that is a fully-fledged PAC learner (has expected loss $\tilde{O}(1/\sqrt{m})$) and is also quite KL-stable (the expected KL is logarithmic in $m$).
> > > >
> > > > Our novel boosting algorithm differs from existing algorithms in that it does not only boost the _accuracy_ of the prediction (reducing the population loss), but also increases the _KL-stability_ (from finite to logarithmic).
> > > >
> > > > A brief sketch of the proof establishing these claims appears in Section 2.2.1, and a complete mathematical proof appears in Appendix A. Intuitively, our algorithm $A^*$ operates by pitting the well-known Multiplicative Weights (MW) and the provided “weak” algorithm $A$ against each other, and using the guarantees of each. More fully, it repeatedly:
> > > >  * Uses the MW algorithm to find a distribution over the samples of the training set on which the hypothesis chosen by the provided “weak” algorithm $A$ is likely to perform poorly.
> > > >  * Uses the “weak” algorithm $A$ to find hypotheses that perform slightly better than random guessing on the distribution chosen by the MW algorithm.
> > > >
> > > > We then use the online regret bound for the MW algorithm to argue that after a modest number of iterations $T = O(\log(m)/\gamma^2)$, there does not exist any sample in the training set that is misclassified by a majority of the hypotheses selected by the “weak” algorithm $A$. Because the the “weak” algorithm $A$ has bounded KL divergence, and it was invoked only $T$ times, a combination of the data-processing inequality and the chain for Rényi divergence imply that the aggregate hypothesis produced by $A^*$ will have a KL divergence proportional to $T$, which is logarithmic in the number $m$ of samples. This also implies generalization.
> > > >
> > > > **Tightness of Our Result**
> > > >
> > > > We agree that it would be beneficial if future experimental works evaluate our algorithm empirically. However, there are already strong reasons to believe that our algorithm does indeed perform well in practice:
> > > >  * The accuracy (low population loss) guarantee for our algorithm follows (as described above) from well-understood techniques in the boosting literature that are known to give good results in practice.
> > > >  * For the bound on the KL-stability, which is novel, notice that our bound is an explicit, non-asymptotic numerical bound:
> > > > $\mathrm{KL}(A^*(S) \| \mathcal{P}^{\star}) \leq T \cdot 2b / \gamma + \log(\pi^2/6 \cdot T^2)$
> > > > (Eq. (9) and line 330 in the Supplementary Materials).
> > > > Seeing as there are no hidden constants that could potentially be large (in Big-O notation etc.), we are confident that our theorem provides a meaningful and straightforward upper bound on the KL divergence of $A^*$. Note also that our result joins a rich literature of _theoretical_ works on boosting (see [4] for an extensive bibliography).
> > > >
> > > > Does this address your concern? If there are any specific remaining concerns, please let us know!

---

> > > > > ### Author Response · Authors · 2023-08-14
> > > > > **References Mentioned in Previous Comment**
> > > > >
> > > > >
> > > > > [1] Schapire, R.E., 1990. The strength of weak learnability. Machine learning, 5, pp.197-227.
> > > > >
> > > > > [2] Freund, Y., 1995. Boosting a weak learning algorithm by majority. Information and computation, 121(2), pp.256-285.
> > > > >
> > > > > [3] Freund, Y. and Schapire, R.E., 1997. A decision-theoretic generalization of on-line learning and an application to boosting. Journal of computer and system sciences, 55(1), pp.119-139.
> > > > >
> > > > > [4] Schapire, R.E. and Freund, Y., 2013. Boosting: Foundations and algorithms. Kybernetes, 42(1), pp.164-166.
> > > > >
> > > > > [5] Shalev-Shwartz, S. and Ben-David, S., 2014. Understanding machine learning: From theory to algorithms. Cambridge university press.
> > > > >
> > > > > [6] Mohri, M., Rostamizadeh, A. and Talwalkar, A., 2018. Foundations of machine learning. MIT press.

---

> > > > > > ### Author Response · Authors · 2023-08-14
> > > > > > **Answer to The Second Question**
> > > > > >
> > > > > > **In addition to the insights presented in Theorem 2.2, could you highlight any other innovative contributions that distinguish your work from the prior literature as listed in Theorem 1.3?**
> > > > > >
> > > > > > Absolutely! Besides our contribution in Theorem 2.2 (stability boosting), there are two additional main contributions in this paper:
> > > > > >
> > > > > > **Contribution I**
> > > > > >
> > > > > > In Theorem 2.1 (which is restated more fully as Theorem B.1 in the supplementary materials), we show an equivalence between:
> > > > > >
> > > > > > (i) Pure Differential Privacy
> > > > > >
> > > > > > (ii) KL-Stability
> > > > > >
> > > > > > (iii) Pure Perfect Generalization
> > > > > >
> > > > > > (iv) $D_\alpha$-Stability with a logarithmic bound $\mathrm{D}_\alpha(A(S) \| \mathcal{P}) \leq O(\log m)$, and
> > > > > >
> > > > > > (v) Finite fractional clique dimension.
> > > > > >
> > > > > > Except for (i) $\iff$ (v), all of these equivalences are **_novel contributions_** introduced in this paper.
> > > > > >
> > > > > > Note that these notions of stability are important and well-motivated: Differential Privacy is arguably one of the most-studied topics in learning theory in the past two decades and has been implemented in practice in many contexts (see e.g. [1]); KL-Stability corresponds to learning using the celebrated PAC-Bayes bound [2] that has been very influential, including more recently in the context of deep learning [3]. Finally, Pure Perfect Generalization is a natural variant of Perfect Generalization, a notion studied in a sequence of recent papers (e.g., [4], [5] and others). Therefore, we believe that showing an equivalence between them is a significant contribution, allowing researchers to use tools and results from one topic to answer questions concerning the other topics.
> > > > > >
> > > > > > In our answer to Reviewer RbiN we mention some examples of these types of connections:
> > > > > > > recall that thresholds over the real line cannot be learned by a differentially private learner. Hence, by Theorem 2.1, there does not exist a PAC learner for thresholds that is KL-stable. Another example is half-spaces with margins in $\mathbb{R}^d$. Half-spaces with margins are differentially private learnable [6], therefore there exists a PAC learner for half-spaces with margins that is KL-stable. Those are examples of the type of connections that our paper elucidates.
> > > > > >
> > > > > > These are just two examples – we are happy to provide many more if you like.
> > > > > >
> > > > > > We view this as a central mathematical contribution of our paper.
> > > > > >
> > > > > > **Contribution II**
> > > > > >
> > > > > > Our paper also makes a conceptual contribution by introducing the notions of distribution-dependent and distribution-independent stability. Each of these two categories contains interesting and important definitions of stability. Noting the similarity within each category and the difference between the categories leads to our observation that the definitions within each category tend to be equivalent to each other, and to be non-equivalent to definitions from the other category. This observation sheds new light on existing work, and helps organize the crowded field of stability definitions into a coherent picture. We believe this too is a worthwhile contribution of our paper.
> > > > > >
> > > > > > _______
> > > > > >
> > > > > > [1] Dwork, C. and Roth, A., 2014. “The algorithmic foundations of differential privacy”. Foundations and Trends in Theoretical Computer Science, 9(3–4), pp.211-407.
> > > > > >
> > > > > > [2] David A. McAllester, "Some PAC-Bayesian Theorems", COLT 1998.
> > > > > >
> > > > > > [3] Gintare Karolina Dziugaite and Daniel M. Roy, "Computing nonvacuous generalization bounds for deep (stochastic) neural networks with many more parameters than training data”, UAI 2017.
> > > > > >
> > > > > > [4] Cummings, Rachel, Katrina Ligett, Kobbi Nissim, Aaron Roth, and Zhiwei Steven Wu. "Adaptive learning with robust generalization guarantees." COLT 2016.
> > > > > >
> > > > > > [5] Bun, Mark, Marco Gaboardi, Max Hopkins, Russell Impagliazzo, Rex Lei, Toniann Pitassi, Satchit Sivakumar, and Jessica Sorrell. "Stability Is Stable: Connections between Replicability, Privacy, and Adaptive Generalization." STOC 2023.
> > > > > >
> > > > > > [6] Blum, A., Dwork, C., McSherry, F., & Nissim, K. (2005, June). Practical privacy: the SuLQ framework. In Proceedings of the twenty-fourth ACM SIGMOD-SIGACT-SIGART symposium on Principles of database systems (pp. 128-138).

---

> > > > > > > ### Comment · Reviewer_xEWt · 2023-08-15
> > > > > > >
> > > > > > > Thank you for your detailed comments. I would like to change my rating from 3 to 5. The main weakness is the poor English writing.

---

> > > > > > > > ### Author Response · Authors · 2023-08-15
> > > > > > > >
> > > > > > > > Thank you! We are thrilled to hear that you are increasing your rating! :)

---

### Official Review · Reviewer_V99Z · 2023-07-03

**Soundness:** 4 excellent
**Presentation:** 4 excellent
**Contribution:** 3 good
**Rating:** 8
**Confidence:** 2

**Summary:**

The paper is a thorough collection of relations between different notions of stability used in learning theory literature. The authors propose a systematization of the many relations by proposing a bifurcation of the notions into two classes -- distribution-independent stability and distribution-dependent stability, where the distribution here refers to population distribution.

**Strengths:**

The paper is a very thorough organization of the panoply of notions of stability used in learning theory literature and provides an insightful perspective on how to view them in relation to each other.

**Weaknesses:**

The amount of content in the paper is better suited for a journal rather than a mere 8 page conference paper. I found the order of the sections to be weird -- section 3 on preliminaries should be before section 2 for example.

**Questions:**

N/A

---

> ### Author Rebuttal · Authors · 2023-08-09
>
> **I found the order of the sections to be weird -- section 3 on preliminaries should be before section 2 for example.**
>
> The current order was chosen so that the main contributions of the paper will appear as early as possible (this is common practice in some conferences). However, we agree that it makes sense to have the preliminaries appear before the technical overview, and we are happy to swap the order of these sections.

---

> > ### Comment · Reviewer_V99Z · 2023-08-21
> >
> > Thank you for the response. I believe once the paper's exposition is improved this paper is a strong accept.

---

### Official Review · Reviewer_RbiN · 2023-07-06

**Soundness:** 2 fair
**Presentation:** 1 poor
**Contribution:** 2 fair
**Rating:** 5
**Confidence:** 3

**Summary:**

This paper categories different definitions of stability (approximate/pure DP, replicability, global stability, perfect generalization, TV indistinguishability, mutual information stability and KL divergence stability) in the literature into two families (distribution-dependent stability and distribution-independent Bayesian stability) and provides theoretical equivalences in both groups.

**Strengths:**

This paper provides connections between a series of stability definitions in the literature and proved the equivalence between them, which has potential to generalize new proof techniques between different topics.

**Weaknesses:**

1. This paper introduces a series of definitions without any literature, for example, $D_\alpha$-Stability, etc. Without any backgrounds/intuitions on the definitions and any future direction section, I am not convinced enough on how there results can be applied to other topics. Some discussions and intuitions for the definitions can greatly improve the quality of the paper.

2. The equivalence results in this paper are useful and I expect to see some examples on how the equivalence results can help building bridges between different stability literature.

3. The writing and structure of the paper can be improved.

**Questions:**

Can the authors provide some examples on what the equivalence results can help connecting the literature of different stability literature?

**Limitations:**

The current version of the paper can be improved by some rigorous introductions on the different definitions of stability introduced in this paper, for example, how these concepts have been applied in different topics, but their connection has rarely been investigated. The current version is more like a list of definitions and theorems, without any intuitions, discussions and future directions. Polishing the paper would greatly strengthen the paper, but given the limitations above, incorporating these into the current paper will require a fair amount of rewriting and editing.

---

> ### Author Rebuttal · Authors · 2023-08-09
>
> **This paper introduces a series of definitions without any literature, for example, $\mathsf{D}_\alpha$-Stability, etc. Without any backgrounds/intuitions on the definitions and any future direction section, I am not convinced enough on how there results can be applied to other topics. Some discussions and intuitions for the definitions can greatly improve the quality of the paper.**
>
> $D_\alpha$ is well motivated. $D_1$ corresponds to KL-divergence and $D_\infty$ corresponds to Perfect Generalization -- both of which are well-studied notions. Therefore, it is natural to consider $D_\alpha$ stability for $\alpha \in (1,\infty)$.
>
> Moreover, there is also literature regarding $D_\alpha$-Stability as well. For example, [1] showed a generalization bound using R\'{e}nyi divergence (Theorem 2 and Corollary 5).
>
> **The equivalence results in this paper are useful and I expect to see some examples on how the equivalence results can help building bridges between different stability literature.**
>
> One example is the connection between pure differential privacy and the PAC-Bayes theorem. Both of these are fundamental ideas that have been extensively studied. Theorem 2.1 states that a hypothesis class admits a pure differentially private PAC learner if and only if it admits a distribution independent KL-stable PAC learner. This is an interesting and non-trivial connection between two well studied notions.
>
> As a concrete example of this connection, recall that thresholds over the real line cannot be learned by a differentially private learner. Hence, by Theorem 2.1, there does not exist a PAC learner for thresholds that is KL-stable. Another example is half-spaces with margins in $\mathbb{R}^d$. Half-spaces with margins are differentially private learnable [2], therefore there exists a PAC learner for half-spaces with margins that is KL-stable. Those are examples of the type of connections that our paper elucidates. (There are many many other possible examples.)
>
>
> **The current version of the paper can be improved by some rigorous introductions on the different definitions of stability introduced in this paper, for example, how these concepts have been applied in different topics, but their connection has rarely been investigated.**
>
> We will add further discussion and background about the definitions of stability that we study.
>
>
>
> **References**
>
> [1] Amedeo Roberto Esposito, Michael Gastpar, and Ibrahim Issa. Robust
> generalization via $\alpha$-mutual information. CoRR, abs/2001.06399,
> 2020.
>
> [2] Blum, A., Dwork, C., McSherry, F., & Nissim, K. (2005, June). Practical privacy: the SuLQ framework. In Proceedings of the twenty-fourth ACM SIGMOD-SIGACT-SIGART symposium on Principles of database systems (pp. 128-138).

---

> > ### Comment · Reviewer_RbiN · 2023-08-14
> >
> > Thank you for your response. I think adding some discussions on the literature and intuitions of the results can greatly improve the paper. The structure and writing of the paper can also be improved.

---

> > > ### Author Response · Authors · 2023-08-18
> > >
> > > We are very happy to add additional discussion on the literature, and on the intuition behind our results.
> > >
> > > Additionally, your review mentioned the following points:
> > >
> > > * **Literature for $D_\alpha$.** We feel we have fully addressed this point in our rebuttal.
> > >
> > > * **Examples on how the equivalence results can be used.** We have provided a number of example in our rebuttal, which we will also include in the final version of the paper.
> > >
> > > Thus, we feel we have fully addressed all the concerns raised in the review. Additionally, all the points raised have touched on non-major issues of presentation and style, and did not find any flaws with the actual contents of our mathematical and conceptual contributions.
> > >
> > > At this point, do you have any substantial concerns regarding our paper? If not, would you be willing to reconsider your score for our paper?

---

> > > > ### Comment · Reviewer_RbiN · 2023-08-20
> > > >
> > > > Thank you for the authors' response. After carefully reading all the reviews and responses, I believe the authors have addressed most of my questions and concerns, I am happy to update my score after I further understand the paper. I think improving the structure and adding more discussions and intuitions can further improve the paper.

---

> > > > > ### Author Response · Authors · 2023-08-20
> > > > >
> > > > > That is wonderful! We are thrilled to see that you have raised your score :)
> > > > >
> > > > > Following your suggestions, in the final version of the paper we will add additional discussion and intuition concerning the definitions of stability appearing in the paper as well as our results and how they can be applied.
> > > > >
> > > > > If there is any specific topic or issue that you find in particular to be not entirely clear, please let us know and we will try to address it here before the end of the author-reviewer discussion :)

---

### Official Review · Reviewer_uUZW · 2023-07-08

**Soundness:** 3 good
**Presentation:** 2 fair
**Contribution:** 3 good
**Rating:** 5
**Confidence:** 2

**Summary:**

**Post-rebuttal**

I thank the authors for their response. I will keep my score as is.


This work aims at building a comprehensive taxonomy of stability definitions showing that many definitions of stability are equivalent to each other.

**Strengths:**

This paper studies the interrelations between different types of algorithmic stability. The goal is to extend the study of equivalences between different notions of stability, and make it more systematic. The main contribution of the paper is to show equivalences between distribution-independent Bayesian notions of stability. The authors also provide a boosting result that enables stability amplification.

**Weaknesses:**

* The paper is very dense. It requires familiarity with a number of concepts related to different notions of stability and also substantial background knowledge about related prior work (all quite recent). Of course since the authors aim at creating a comprehensive taxonomy of stability definitions, this is expected and the paper does a great job at that, both in relation to prior work for the distribution-dependent equivalences, and the new ones for the distribution-independent versions. I could not devote enough time to go through the supplied proofs but overall, am not entirely convinced if a conference venue like NeurIPS is the ideal platform for such a dense and terse presentation. The presentation can perhaps be improved in certain parts to improve readability, e.g., giving more intuition wherever applicable (the current draft does not saturate the 9 page limit).


* TV indistinguishability (alluded to in the abstract) is not defined in the main text or supplementary.

**Questions:**

* How is TV indistinguishability related to TV stability?

**Limitations:**

I do not foresee any potential negative societal impact of this work.

---

> ### Author Rebuttal · Authors · 2023-08-09
>
>  **TV indistinguishability (alluded to in the abstract) is not defined in the main text or supplementary. How is TV indistinguishability related to TV stability?**
>
>
> They are the same. [1] used the term TV-indistinguishability and we preferred to use the term TV-stability, we believe that it emphasises the fact that this definition is a form of algorithmic stability. We do note that [1] defined TV-indistinguishability slightly different from our definition of TV-stability (see Definition 4 in [1]), but they also showed in Appendix A.3.1 that their definition is equivalent to ours (up to some constant factors).
> The term "TV-indistinguishability'' appears only once (in the abstract), we will change this to "TV-stability'' to eliminate this potential confusion.
>
> **The presentation can perhaps be improved in certain parts to improve readability, e.g., giving more intuition wherever applicable (the current draft does not saturate the 9 page limit).**
>
>  We will add more background and intuition about the various notions of stability studied in this paper and the connections between them.
>
>
> [1] Alkis Kalavasis, Amin Karbasi, Shay Moran, and Grigoris Velegkas.
> Statistical indistinguishability of learning algorithms, 2023.

---

### Official Review · Reviewer_c5ne · 2023-08-02

**Soundness:** 3 good
**Presentation:** 3 good
**Contribution:** 3 good
**Rating:** 7
**Confidence:** 1

**Summary:**

The paper shows that many definitions of the "stability" in learning theory are equivalent (according to a specific definition of equivalence).
They prove this is the case for various distribution-independent definitions of stability as well as compile the same result for distribution-dependent definitions from previous work.
This summary allows a unified view across many stability-related concepts in the literature.
The submission also proves a result related to boosting, showing that a weak learner can be amplified to obtain better stability and learning performance.

**Strengths:**

The paper is well-written and has a clear organization.
The concept of stability studied in the paper is an important one in learning theory.
By showing that multiple definitions of stability are weakly equivalent, the authors obtain important results that allow these definitions to be related to each other in a coherent manner.

**Weaknesses:**

I don't have many complaints about this work.
I suggest the authors consider clarifying the details asked in my questions in the following section to make the paper clearer.

**Questions:**

- I'm having some trouble understanding the motivation behind wanting the difference between the prior and the posterior to be small in the description of stability on lines 33–34.
It seems to me we actually want the difference between two possible posteriors when applied to two similar realizations of $S$ to be small.
What if the data really disagree with the prior; wouldn't we have a large difference between the prior and the posterior regardless?
- On line 39, what makes the definition of stability Bayesian exactly? Because we iterate over possible realizations of $S$?
- In Section 2.1.1, does the chain mean only Pure DP implies $\text{D}_\infty$-Stability and the like but not vice versa ($\text{D}_\infty$-Stability implying Pure DP)?

**Limitations:**

The authors did not discuss any possible limitations of their results, although it might be the case that the discussion doesn't apply due to the nature of the topic.

---

> ### Author Rebuttal · Authors · 2023-08-09
>
> **I'm having some trouble understanding the motivation behind wanting the difference between the prior and the posterior to be small in the description of stability on lines 33–34.**
>
> Considering the "distance" between a prior distribution and the posterior is very common. (The word "distance" is in quotation since the measure of dissimilarity between them does not need to be a metric, e.g. the Kullback–Leibler divergence.)
> For example, in the context of generalization, PAC Bayes Theorem assures that for every population distribution and any given prior $\mathcal{P}$, the difference between the population error of an algorithm $A$ and the empirical error is bounded by $\tilde{O}\left(\frac{\sqrt{\mathtt{KL}(A(S),\mathcal{P})}}{m}\right)$, where $A(S)$ is the posterior distribution, $\mathtt{KL}(A(S),\mathcal{P})$ is the KL divergence between the prior and the posterior (``measure of dissimilarity"), and $m$ is the size of the input sample $S$. See e.g. Theorem 31.1 in [1].
>
> **It seems to me we actually want the difference between two possible posteriors when applied to two similar realizations of $S$ to be small. What if the data really disagree with the prior; wouldn't we have a large difference between the prior and the posterior regardless?**
>
> Your suggestion of measuring the dissimilarity between two posterior distributions is widely used as well, this is very related to the definition of Replicability. In the paper we proved that those notions of stability are equivalent.
>
> **On line 39, what makes the definition of stability Bayesian exactly? Because we iterate over possible realizations of $S$?**
>
> Our choice of the name *Bayesian* stability is inspired by Bayesian statistics, which uses the terms *prior* and *posterior*. In Bayesian statistics the analyst has some prior distribution over possible hypothesis before conducting the analysis, and chooses a posterior distribution over hypotheses when the analysis is complete. Bayesian stability is defined in terms of the dissimilarity between these two distributions.
> We will happily add this remark to the final version of the paper to clarify our choice of the term `Bayesian stability'.
>
> **In Section 2.1.1, does the chain mean only Pure DP implies $D_\infty$-Stability and the like but not vice versa ($D_\infty$-Stability implying Pure DP)?**
>
> No, $D_\infty$-stability does imply pure DP, since $D_\infty$-stability implies $D_\alpha$-stability for every $\alpha\in[1,\infty]$, which in turn implies pure DP.
>
>
> [1] Shai Shalev-Shwartz and Shai Ben-David. Understanding Machine
> Learning– From Theory to Algorithms. Cambridge University
> Press, 2014.

---

> > ### Comment · Reviewer_c5ne · 2023-08-10
> > **thanks**
> >
> > I thank the authors for their response. I will keep my score as is.

---

### Decision · Program_Chairs · 2023-09-21

**Decision:**

Accept (poster)

**Comment:**

This manuscript concerns various notions of "stability" in Bayesian procedures that have appeared in the literature and establishes some novel equivalences among this "zoo" of possible choices.

Overall, the reviewers felt this topic was compelling and would be of interest to the NeurIPS community. No major objections were raised during the review or discussion period.

However, the discussion period did prove fruitful in revealing and clarifying a number of confusing elements in the manuscript as submitted. I strongly suggest that the authors carefully reflect on this feedback in preparing an updated version of their manuscript.